# Resolvin T4 enhances macrophage cholesterol efflux to reduce vascular disease

Mary E. Walker[1,3], Roberta De Matteis [1,3], Mauro Perretti [1,2] & Jesmond Dalli [1,2] ✉

While cardiovascular disease (CVD) is one of the major co-morbidities in patients with rheumatoid arthritis (RA), the mechanism(s) that contribute to CVD in patients with RA remain to be fully elucidated. Herein, we observe that plasma concentrations of 13-series resolvin (RvT)4 negatively correlate with vascular lipid load in mouse inflammatory arthritis. Administration of RvT4 to male arthritic mice fed an atherogenic diet significantly reduces atherosclerosis. Assessment of the mechanisms elicited by this mediator demonstrates that RvT4 activates cholesterol efflux in lipid laden macrophages via a Scavenger Receptor class B type 1 (SR-BI)-Neutral Cholesterol Ester Hydrolase-dependent pathway. This leads to the reprogramming of lipid laden macrophages yielding tissue protection. Pharmacological inhibition or knockdown of macrophage SR-BI reverses the vasculo-protective activities of RvT4 in vitro and in male mice in vivo. Together these findings elucidate a RvT4-SR-BI centered mechanism that orchestrates macrophage responses to limit atherosclerosis during inflammatory arthritis.

Chronic inflammatory conditions are characterized by multi-morbidities that further aggravate disease burden. In many cases these are linked with a reduced quality of life and increased mortality[1–3]. Patients with rheumatoid arthritis (RA) are at an increased risk of developing cardiovascular disease including atherosclerosis[4]. Atherosclerosis is characterized by the accumulation of lipids, macrophages and fibrotic elements in the intima of large arteries at sites of low or disturbed flow[5–7]. This accumulation results in the formation of atherosclerotic plaques that in most cases are silent with limited clinical consequence. However, in some instances, altered macrophage responses with ensuing excessive lipid accumulation and defective efferocytosis lead to the formation of a necrotic core and a thin fibrous cap. This results in unstable plaques that rupture, causing ischemic cardiovascular disease and even death[5,6]. Emerging evidence indicates that failure to engage the production and/or biological actions of protective mediators that counter-regulate the formation of pro-inflammatory molecules is a common mechanism in the onset and development of chronic inflammatory conditions[5,8–14].

Amongst these protective molecules, recent studies uncovered a role for a genus of mediators termed specialized pro-resolving mediators (SPM)[12]. These molecules are produced via the stereoselective conversion of the omega-6 fatty acid arachidonic acid and the omega-3 fatty acids n-3 docosapentaenoic acid, eicosapentaenoic acid and docosahexaenoic acid, and exert potent stereospecific actions. We found that n-3 docosapentaenoic acid is converted to an SPM family, termed 13-series resolvins (RvTs) that exert vascular-directed actions[15,16]. In the formation of these mediators, n-3 docosapentaenoic acid is converted via cyclooxygenase (COX)−2 to 13-hydroperoxydocosapentaenoic acid, that in turn is converted via a lipoxygenase (ALOX) reaction to four structurally distinct mediators: RvT1, RvT2, RvT3 and RvT4. In infections, RvTs counter-regulate the production of both inflammatory cytokines, such as interleukin-1β, and eicosanoids. Statins upregulate RvTs formation, and inhibition of their biosynthetic pathways reverses the protective actions of these drugs in infectious inflammation and inflammatory arthritis[15,16]. The potential relevance of RvTs in regulating vascular disease is supported by findings made in recent studies where

[1]William Harvey Research Institute, Faculty of Medicine and Dentistry, Queen Mary University of London, Charterhouse Square, London EC1M 6BQ, UK. [2]Centre for Inflammation and Therapeutic Innovation, Queen Mary University of London, London, UK. [3]These authors contributed equally: Mary E. Walker, Roberta De Matteis. ✉e-mail: j.dalli@qmul.ac.uk

administration of highly purified eicosapentaenoic acid, the precursor of n-3 docosapentaenoic acid, together with statins reduced incidence of ischemic events in patients with high triglycerides[17].

Macrophages are central in the pathogenesis of both RA and atherosclerosis[5,6,14,18]. Monocytes are recruited into the inflamed tissues where they differentiate into macrophages. During self-limited inflammation, these cells give rise to resolution-phase macrophages, which are involved in both termination of inflammation and initiation of repair and regenerative processes[19,20]. In disease, such as RA and atherosclerosis, the inflammatory microenvironment can alter the behavior of these cells, promoting an inflammatory phenotype that propagates local inflammation and leads to tissue destruction. In atherosclerosis these cells become lipid laden, with the formation of distinct lipid droplets in their cytoplasm giving rise to foam cells[5,6,14]. This lipid enrichment contributes to an increase in plasma cholesterol resulting in enhanced inflammatory signaling through clustering-mediated activation of signaling receptors. This increased signaling is in turn linked with a propagation of local inflammation[21,22]. Thus, elucidation of mechanisms that regulate lipid-droplet accumulation in macrophages may identify tractable targets for the treatment of vascular inflammation and potentially also joint inflammation in RA.

Given the regulatory actions exerted by RvT in controlling macrophage responses and the protection displayed by omega-3 fatty acids and statin treatment in both RA and atherosclerosis[16,17,23], we questioned whether disruption(s) in the RvT pathways/activity may contribute to vascular inflammation in inflammatory arthritis. In this study we find that plasma RvT concentrations are negatively correlated with vascular lipid load. Furthermore, administration of RvT4 to arthritic mice fed an atherogenic diet leads to a reduction in vascular inflammation via the regulation of macrophage lipid load.

## Results

### Plasma RvT concentrations are negatively correlated with joint and vascular inflammation

To investigate the potential role of RvT in the etiopathogenesis of atherosclerosis in RA we first modeled the relationship between vascular disease and RvT concentrations in inflammatory arthritis. For this purpose, mice were fed either a standard chow diet or a western diet (WD) and subjected to prolonged K/BxN serum-induced arthritis. In mice given WD we found that joint inflammation was increased, as measured by an increase in clinical score throughout the course of the experiment (Fig. 1a) in accord with published findings[24]. This macroscopic outcome was linked to increased joint damage as demonstrated by a reduction in cartilage integrity using safranin O staining (Fig. 1b). In mice fed a WD we also observed an increase in vascular lipid load as determined by an increase in oil-red O staining in the aortic arches of mice fed WD (Fig. 1c), an observation that was in accord with published literature[24]. Assessment of pro-inflammatory eicosanoids in vascular tissues from these mice demonstrated an increase in the levels of these autacoids, including $PGE_2$, in tissues from mice fed a WD when compared with mice fed a standard chow diet (Fig. 1d and Supplementary Fig. 1). We then examined the relationship between systemic RvT concentrations and vascular disease during arthritic inflammation. Here we found that plasma concentrations of RvT1 and RvT4 were reduced in arthritic mice fed a western diet (WD) when compared with mice fed a chow diet, reaching statistical significance for RvT4 (Fig. 1e and Supplementary Fig. 1). We next assessed whether plasma RvTs concentrations were linked with vascular inflammation. Of the two RvTs identified in plasma, we found that concentrations of RvT4 were negatively correlated with the extent of vascular disease in WD-fed mice (Fig. 1f, g and Supplementary Fig. 1).

Since RvTs are produced via the stereoselective conversion of essential fatty acids by specific biosynthetic enzymes, we next queried whether the observed reduction in RvT concentrations may be related to an altered enzymatic activity. To address this, we first investigated

the concentrations of 13-HDPA, the RvT precursor and a marker of COX-2 activity, the initiating enzyme in the RvT pathway[15]. Liquid chromatography-tandem mass spectrometric analysis of 13-HDPA in plasma from chow fed mice and WD-fed mice demonstrated a significant downregulation of 13-HDPA (Supplementary Fig. 2). We next assessed the activity of ALOX5, the second enzyme in the RvT pathway, measuring its n-3 DPA product 7-HDPA. Here we found that 7-HDPA concentrations were also reduced in plasma from WD-fed mice when compared with chow fed mice (Supplementary Fig. 2). Together these findings establish a link between blood RvT concentrations and degree of vascular disease in inflammatory arthritis. Thus, we next assessed the potential mechanisms activated by RvT4 to exert vascular protection in arthritis.

### RvT4 exerts vascular protective actions regulating tissue macrophage responses in inflammatory arthritis

RvT4 was prepared using biogenic synthesis[15] (Supplementary Fig. 3) and administered to arthritic mice fed WD, 8 weeks after WD administration and 4 weeks after the onset of arthritis (see Fig. 2a). Using oil-red O stain we observed a significant reduction in vascular lipid load in the aortic arch of arthritic mice fed a WD and treated with RvT4 (Fig. 2b). In aortas harvested from these mice we also found a marked upregulation of a panel of genes associated with vascular protection. The upregulation of *MerTK* reached statistical significance when compared with levels observed in mice treated with vehicle alone (Fig. 2c). Assessment of SPM concentrations in these tissues also demonstrated that RvT4 markedly regulated the expression of molecules involved in the resolution of inflammation. Principal component analysis (PCA) unveiled a shift in the SPM concentrations in aortas from mice treated with RvT4 when compared with mice treated with vehicle alone. This was demonstrated by a leftward shift in the cluster representing these mice, when compared with mice receiving vehicle. This separation in mediator profiles was linked with the upregulation of several pro-resolving mediators including RvD3 and $LXB_4$ (Fig. 2d, Supplementary Table 1 and Supplementary Figs. 4 and 5a, b).

MerTK is emerging as a central determinant that regulates vascular macrophage responses favoring an accelerated resolution of vascular inflammation[25]. Given that we found a significant increase in the expression of this gene in aortas from mice treated with RvT4 we next questioned whether this mediator modulated vascular macrophage responses to limit atherosclerotic lesion formation in arthritic mice. To test this, we obtained aortas from arthritic mice fed a WD and incubated them with or without RvT4 for 16 h. Assessment of lipid load in macrophages demonstrated a significant reduction in lipid content when RvT4 was added to these organ cultures, as measured using LipidTOX staining (Fig. 2e). Flow cytometric assessment of phenotypic markers on macrophages isolated from these aortas demonstrated that RvT4 markedly altered aortic macrophage polarization. This was illustrated using PCA, which gave a shift in the cluster representing cells obtained from RvT4-treated tissues when compared with vehicle-treated cells (Fig. 2f, Supplementary Table 2 and Supplementary Figs. 5c, d and 6). This change in macrophage phenotype was linked with an upregulation of MerTK, as well as downregulation in the expression of several activation markers, including CD11c and iNOS (Supplementary Table 2), both determinants that are linked with propagation of atherosclerotic lesions[26,27].

### RvT4 upregulates protective responses in lipid-laden macrophages

To further evaluate whether the changes in macrophage phenotype and lipid load were linked with the observed reduction of vascular inflammation, we next evaluated whether RvT4 regulated macrophage responses central to the resolution of inflammation. One of the biological actions that is disrupted in macrophages during

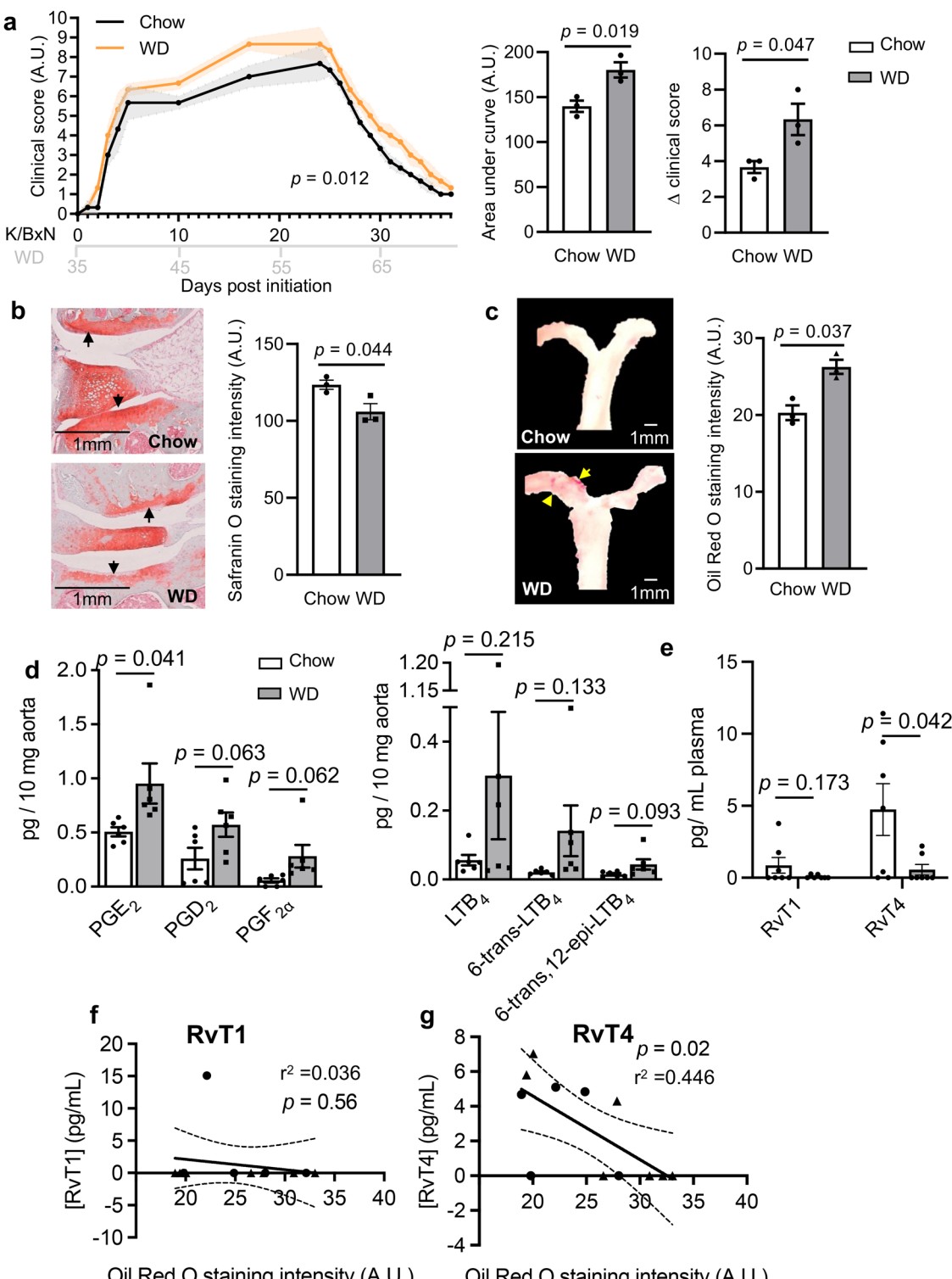

**Fig. 1 | Western-style diet is associated with RvT deficiency and increased disease pathology.** Wild-type C57BL/6 mice were fed chow or Western-style diet. K/BxN serum-induced arthritis was initiated at 11 weeks and sustained by giving three weekly boosters. **a** Disease progression was measured using a 26-point clinical score. $n = 3$ per group; Statistical differences were evaluated using 2-way ANOVA for clinical scores. Mice were culled at 16 weeks, A.U. = arbitrary units. **b** knee joints were collected and Safranin O staining intensity assessed. Mean gray intensity was calculated using ImageJ software. $n = 3$ per group. **c** Aortas were isolated and lipid content assessed using Oil-Red O staining. Aortic lipid content was calculated using ImageJ software. $n = 3$ per group. **d** Lipid mediators were identified and quantified in aortic sections using targeted LC-MS/MS profiling. $n = 6$ per group from two separate experiments. **e** Wild-type C57BL/6 mice were fed

chow or Western-style diet. K/BxN serum-induced arthritis was initiated at 11 weeks. Six days later plasma was collected and, lipid mediators were identified and quantified using targeted LC-MS/MS profiling. $n = 7$ per group from two separate experiments. Statistical differences were evaluated using linear regression. Unless specified, error bars, mean ± s.e.m.; Statistical differences were evaluated using unpaired *t*-test. **f**, **g** WT (circles) and ApoE−/− (triangles) mice with prolonged arthritis were fed a standard chow diet or WD for 11 weeks. Aortas and plasma were harvested and lipid load in the aortas was determined using oil-red O. Plasma RvT1 and RvT4 concentrations were measured using LC-MS/MS profiling. RvT concentrations were then correlated with aortic lipid content. Statistical differences were evaluated using linear regression. Dashed lines = 95% confidence interval. $n = 12$ mice from three separate experiments.

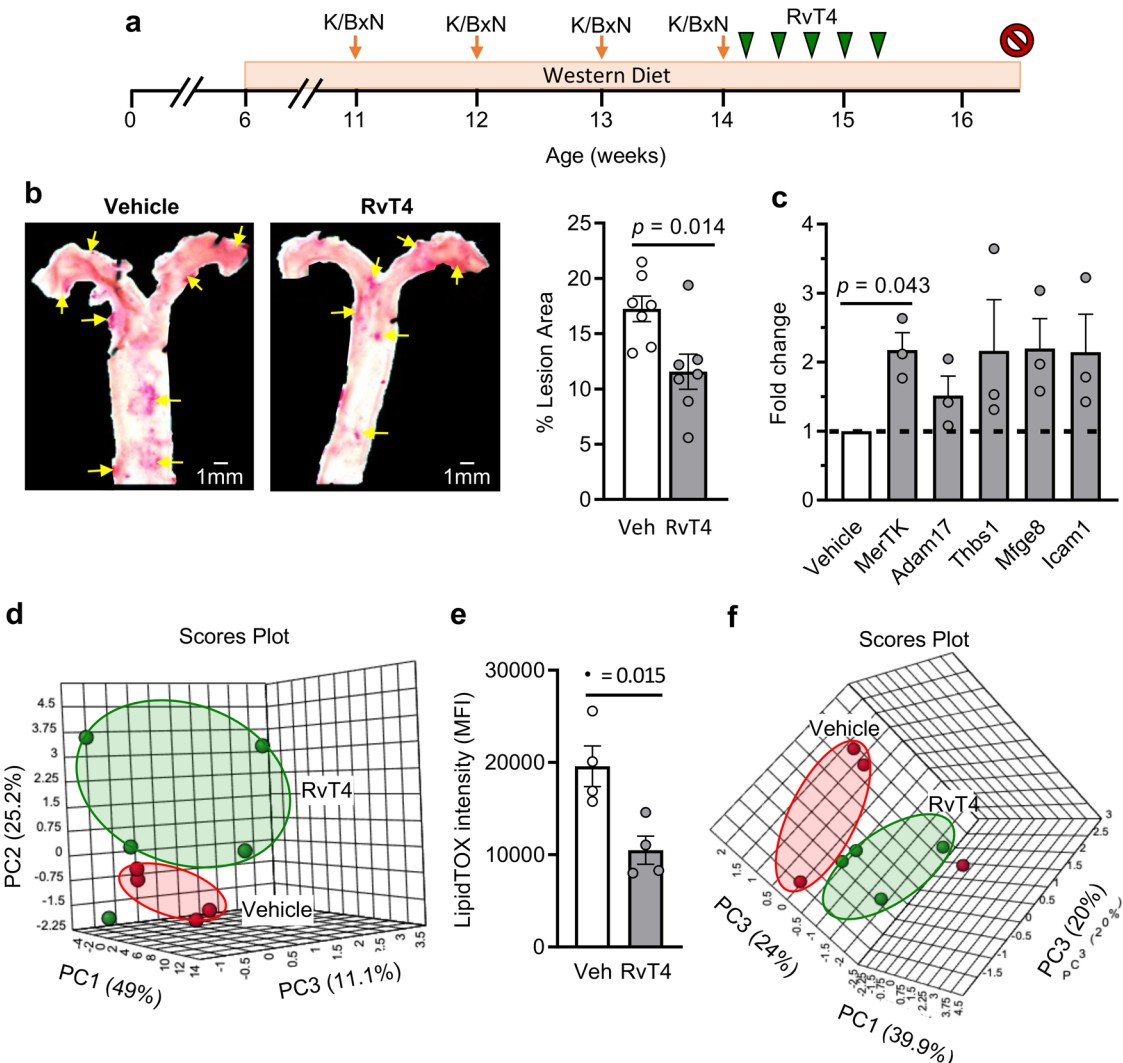

**Fig. 2 | RvT4 reduces vascular inflammation during hyperlipidaemia and arthritis. a** Schematic representation of the experiment: wild-type C57BL/6 mice were fed a Western-style diet and K/BxN serum-induced arthritis sustained for 4 weeks. After the final boost, mice were injected with vehicle or 75 ng RvT4 on alternating days for 10 days. Mice were culled at 16 weeks. **b** Aortas were isolated and vascular inflammation assessed by measuring lesion size. $n = 7$ per group; $p$ value was calculated using unpaired $t$-test. **c** Gene expression of macrophage-associated phenotypic markers in aortas. $n = 3$ per group; Statistical differences were evaluated using one-sample $t$-test. **d** Aortic lipid mediator profiles were measured using targeted LC-MS/MS and assessed using principal component analysis, resulting in a 3D Scores Plot. Input data were SPM in Supplementary

Table 1, and associated loading plot and scree plot in Supplementary Fig. 5. Wild-type mice were fed a Western-style diet and serum-induced arthritis initiated. At day 6 of arthritis, aortas were collected and incubated for 16 h with vehicle (0.03% ethanol) or 1 nM RvT4. Single-cell suspensions were prepared, and **e** lipid content in CD64+ F4/80+ cells were measured using LipidTOX stain. $n = 4$ per group; Statistical differences were evaluated using unpaired $t$-test. **f** The expression of phenotypic markers on macrophages was quantified using fluorescently labeled antibodies and flow cytometry. Principal component analysis of the macrophage markers measured in CD64+ F4/80+ cells, represented by 3D Scores Plot. Input data were fluorescence intensities in Supplementary Table 2. Associated loading plot and scree plot are in Supplementary Fig. 5. Error bars, mean ± s.e.m.

atherosclerosis is efferocytosis[6,14,28]. This altered ability of macrophages to clear dead cells is linked with necrotic core formation in the vascular wall[6,14,28]. Therefore, we next assessed the ability of RvT4 to upregulate macrophage efferocytosis in oxLDL-loaded cells. Assessment of apoptotic cell uptake in real-time using high content imaging demonstrated that the presence of RvT4 increased both the rate at which lipid-laden macrophages engulfed apoptotic cells as well as their overall capacity to ingest these cells (Fig. 3a, b). Thirty minutes after the addition of apoptotic cells, efferocytosis in lipid-laden macrophages incubated with 1 nM of RvT4 was increased by 131% compared to vehicle, whereas efferocytosis in lipid-laden macrophages incubated with 0.1 nM or 10 nM of RvT4 was increased by 54% and 90% respectively when compared to cells incubated with vehicle alone (Fig. 3a).

Lipid-laden macrophages also display an impaired ability to migrate, resulting in these cells becoming trapped in tissues and perpetuating local inflammation[6,14]. Thus, we next assessed whether RvT4 also improved macrophage motility towards chemotactic stimuli. While incubation of lipid-laden macrophages with RvT4 alone did not increase migration in the absence of a chemoattractant (Fig. 3c), cell migration towards two classic chemoattractants, MCP-1 (Fig. 3d) and ATP (Fig. 3e), was increased by addition of RvT4 in a concentration-dependent manner. These results indicate that RvT4 restores the ability of lipid-laden macrophages to respond to a chemotactic queue.

To establish if this increased chemotaxis was observed in the diseased tissues, we incubated aortic arches from arthritic mice fed WD with 1 nM of RvT4. After 16 h, we found a significant increase in the

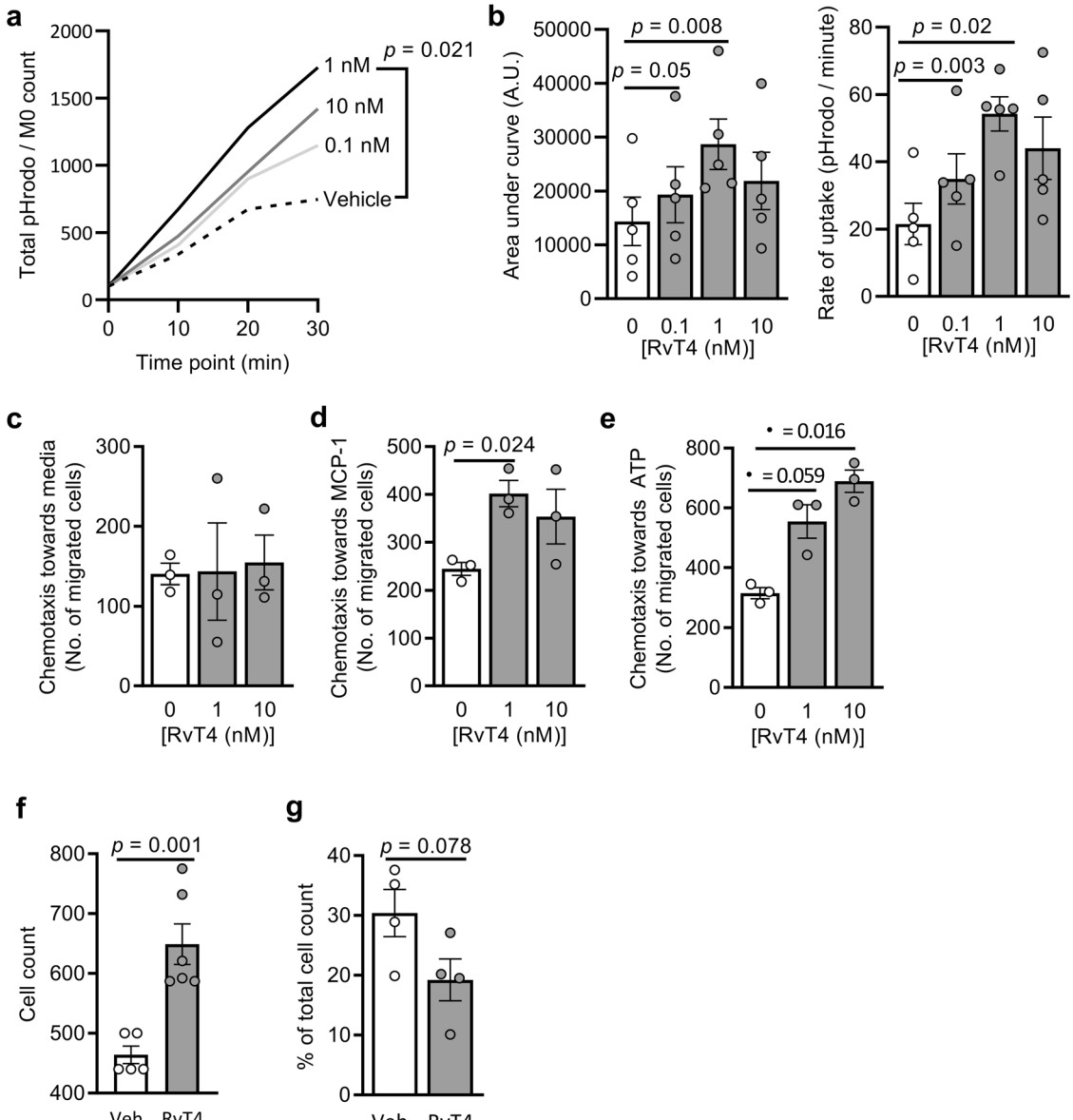

**Fig. 3 | RvT4 increases phagocytosis and chemotaxis. a** oxLDL-loaded monocyte-derived macrophages were incubated for 30 min with vehicle or 1 nM RvT4. Apoptotic HL-60 cells stained with pHrodo SE were then added, and phagocytosis measured. $n = 5$ per group; Statistical differences were evaluated using repeated measures 2-way ANOVA. **b** Area under curve (left) and rate of apoptotic cell uptake (right) were calculated using GraphPad Prism software. Mouse bone marrow-derived oxLDL-loaded macrophages were incubated with vehicle or RvT4 (1 or 10 nM). A.U. = arbitrary units. Chemotaxis was assessed using **c** vehicle, **d** MCP-1 or **e** ATP as chemoattractants, and migrated cells enumerated. $n = 3–4$ per group;

Statistical differences were evaluated using 1-way ANOVA. Aortas of ApoE[-/-] mice fed a Western-style diet for 8 weeks were isolated and incubated with vehicle of RvT4. **f** At the end of the incubation, cells that had migrated out of the tissue onto the well were collected and enumerated. **g** Aortas were collected, cells were liberated and the percentage CD64 + F4/80+ cells in total cell count were determined. $n = 4–5$ per group; Statistical differences were evaluated using unpaired $t$-test. Unless specified, error bars, mean ± s.e.m.; all ANOVA results were corrected for multiple comparisons by controlling the False Discovery Rate using the two-stage step-up method of Benjamini, Krieger and Yekutieli. Results are from two separate experiments.

total number of cells that migrated out of the aortic tissues (Fig. 3f). This was linked with a proportional and significant reduction in tissue macrophages (Fig. 3g). These findings indicate that RvT4-initiated changes in macrophage lipid load were linked with an improvement in the cellular responses required to promote resolution of tissue inflammation. Thus, we next set out to establish the mechanisms by which RvT4 reduced macrophage lipid load.

## RvT4 promotes cholesterol efflux in macrophages
We first determined whether RvT4 directly regulated cholesterol efflux in macrophages, the mechanism responsible for reducing lipid load in macrophages[29,30]. Addition of RvT4 to oxLDL-loaded human

monocyte-derived macrophages rapidly increased the release of cholesterol from macrophages as measured by a significant increase in total cholesterol concentrations in cell supernatants (Fig. 4a). This increase in extracellular cholesterol levels was coupled with a reduction in intracellular lipid load (Fig. 4b).

In the formation of lipid droplets, free cholesterol is esterified into cholesterol esters[29,30]. Having observed an increase in free cholesterol release in macrophages incubated with RvT4, we queried whether this mediator regulated the hydrolysis of cholesterol esters with consequent augmentation of intracellular free cholesterol. Incubation of lipid-laden macrophages with RvT4 led to a rapid increase in intracellular free cholesterol levels, as measured by an increase in the

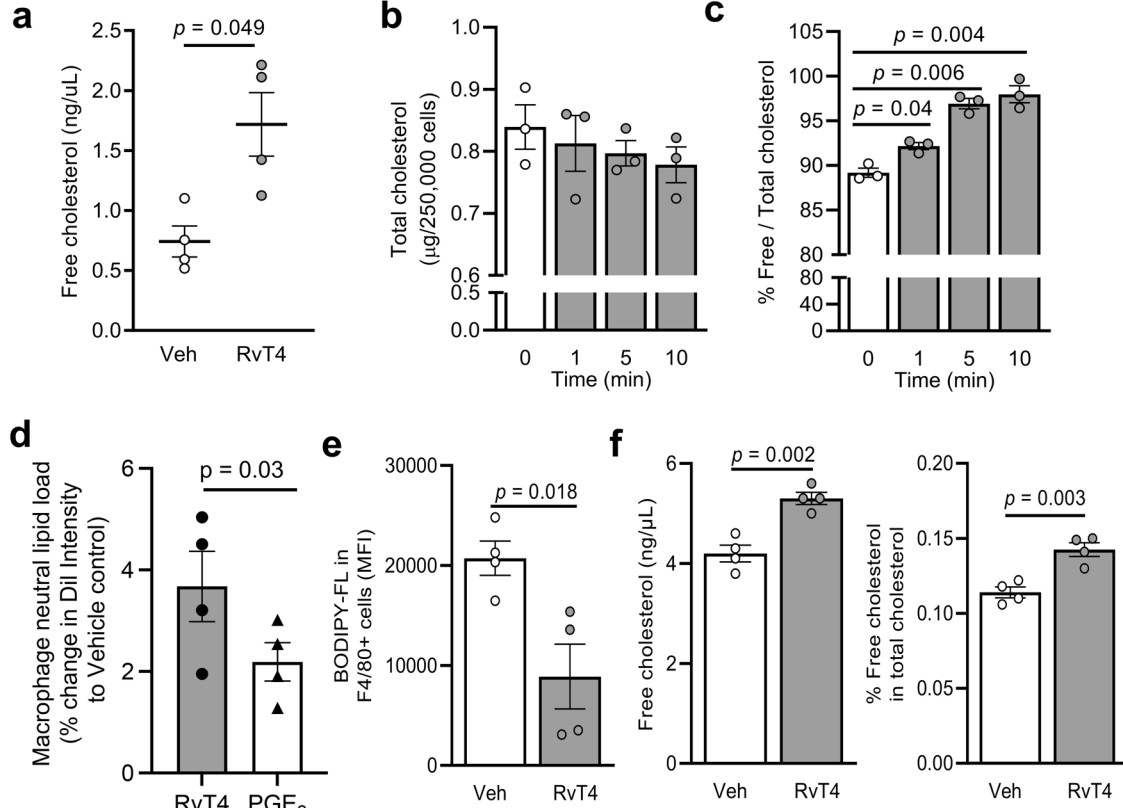

**Fig. 4 | RvT4 promotes cholesterol efflux in macrophages to reduce lipid load.**
Peripheral blood mononuclear cells were isolated from healthy human donors and incubated with GM-CSF over 6 days. **a** oxLDL-loaded macrophages were incubated with vehicle or 1 nM RvT4 for 45 min, and free cholesterol concentration in supernatants were measured. $n = 4$ per group, 2 independent experiments; $p$ value was calculated using unpaired $t$-test. **b**, **c** oxLDL-loaded macrophages were incubated with 1 nM RvT4 for 0, 1, 5 or 10 min, and supernatant removed. Intracellular free and total cholesterol concentrations were measured. $n = 3$ per group, 2 independent experiments; Statistical differences were evaluated using repeated measures 1-way ANOVA and results were corrected for multiple comparisons by controlling the False Discovery Rate using the two-stage step-up method of

Benjamini, Krieger and Yekutieli. **d** Human monocyte-derived macrophages were incubated with DiI-labeled oxLDL for 16 h then with vehicle, RvT4 (1 nM) or PGE$_2$ (1 nM) for 45 min and lipid load evaluated. Error bars, mean ± s.e.m.; $n = 4$ per group, 2 independent experiments. Statistical differences were evaluated using paired $T$-test. **e**, **f** ApoE$^{-/-}$ mice were injected i.p. with 5 μg BODIPY-FL-cholesterol in PBS. After 16 h mice were administered with vehicle (DPBS$^{-/-}$ + 0.01% EtOH) or 75 ng RvT4 via i.p. injection and 2 h later peritoneal lavages were carried out. **e** F4/80+ cells were identified using flow cytometry, and their BODIPY-FL intensity measured. **f** Free and total cholesterol was measured in the lavage fluid; $n = 4$ per group; Statistical differences were evaluated using unpaired $t$-test. Unless specified, error bars, mean ± s.e.m. Results are from two separate experiments.

percentage of free cholesterol when compared with total intracellular cholesterol levels (Fig. 4c). We next evaluated whether this ability of RvT4 to reduce intracellular cholesterol load was shared with other lipid mediators. Incubation of lipid-loaded macrophages with PGE$_2$, a lipid mediator found to be upregulated in mice fed a WD, led to a decrease in intracellular lipid load, although this was significantly less than that induced by RvT4 (Fig. 4d).

We next sought to evaluate whether our in vitro findings were translatable to the in vivo scenario. For this purpose, a fluorescently labeled cholesterol was delivered into the peritoneal cavity of ApoE$^{-/-}$ mice. After 16 h, these animals were administered RvT4 or vehicle and intracellular cholesterol load in peritoneal macrophages was evaluated using flow cytometry. This experiment revealed a significant reduction in intracellular cholesterol levels in cells recovered from the peritoneal cavity of mice treated with RvT4 when compared with mice that received vehicle alone (Fig. 4e and Supplementary Fig. 7). This decrease in intracellular cholesterol was coupled with a significant increase in extracellular cholesterol measured in cell-free peritoneal lavages from RvT4-treated mice (Fig. 4f). Together these findings lend support to the hypothesis that RvT4 regulates cholesterol efflux in macrophages. Thus, we next sought to determine the mechanism(s) activated by RvT4 that mediate this protective response.

## RvT4 upregulates SR-BI expression on the plasma membrane of macrophages to reduce lipid load

Release of cholesterol from neutral lipids is an enzymatically regulated process that in macrophages is primarily catalyzed by two enzymes: lipid-droplet-associated hydrolase (LDAH)[31] and neutral cholesterol esterase hydrolase (nCEH)[32]. To determine their potential involvement, lipid-laden human monocyte-derived macrophages were transfected with siRNA to either of these enzymes (or a non-targeting control siRNA) and then we assessed the ability of RvT4 to reduce macrophage lipid load. Transfection of cells with siRNA targeting either of the enzymes led to a decrease in mRNA encoding for the proteins and increased intracellular lipid load in these macrophages (Fig. 5a). RvT4 retained its ability to reduce macrophage lipid load in macrophages transfected with siRNA against LDAH, while this protective effect was lost in cells transfected with siRNA against nCEH (Fig. 5b).

Hydrolysis of neutral lipids releases free cholesterol which can be toxic to the cell and therefore must be rapidly exported. This process is primarily orchestrated by the ATP-binding cassette transporters (ABCA1 and ABCG1) and Scavenger Receptor B type 1 (SR-BI)[29,30]. Thus, we tested the contribution of each of these pathways to the observed RvT4-mediated decrease in macrophage lipid load. Incubation of oxLDL-loaded macrophages with Cyclosporine A (CycA), an

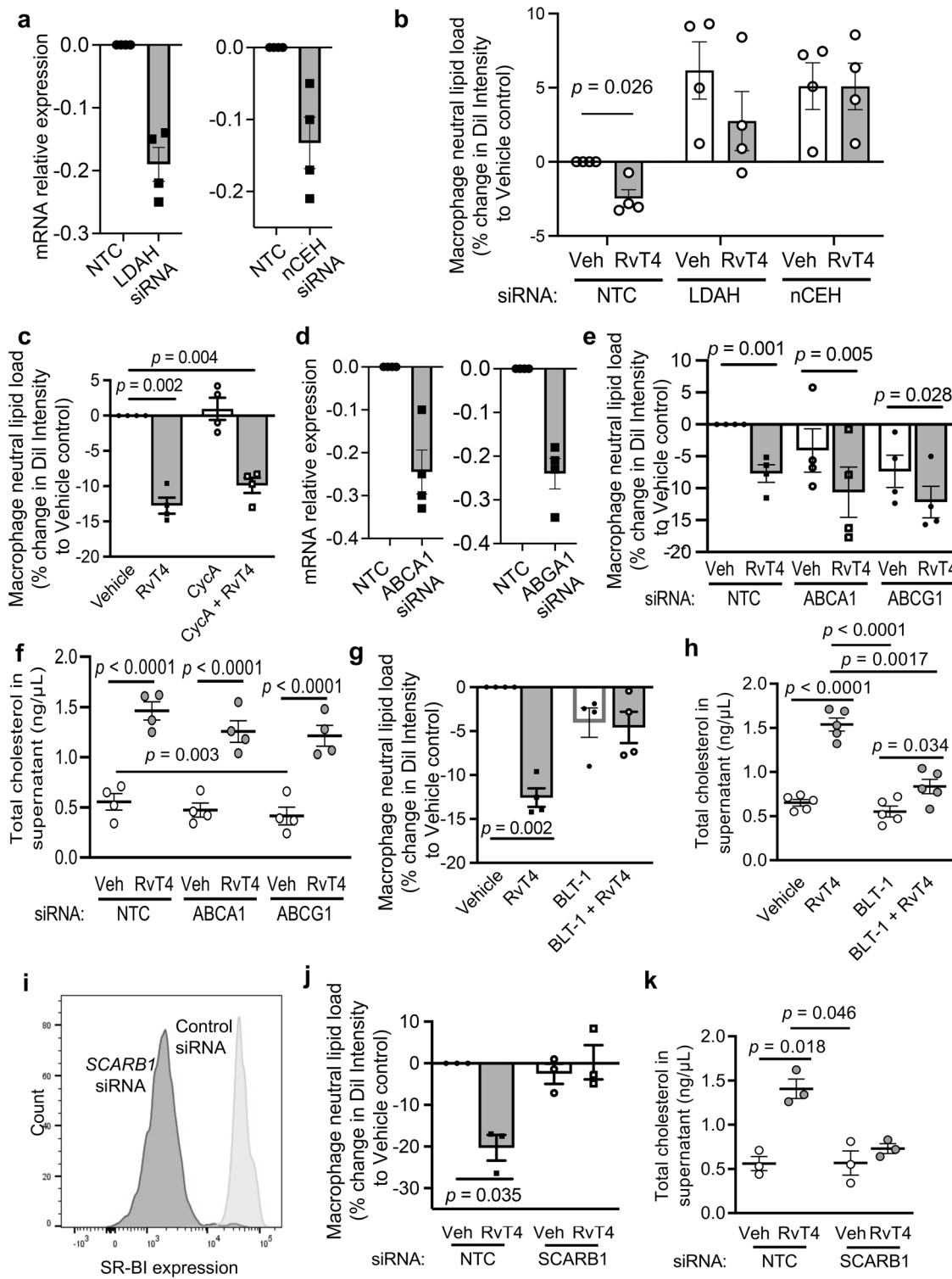

inhibitor to both ABCA1 and ABCG1, did not block the ability of RvT4 to promote cholesterol efflux (Fig. 5c). These findings were further confirmed by knocking down the expression of ABCA1 and ABCG1 using siRNA technologies (Fig. 5d). Indeed, knockdown of ABCA1 or ABCG1 expression did not limit the ability of RvT4 to promote the release of cholesterol from macrophages (Fig. 5e, f). Conversely, incubation of these cells with BLT-1, an inhibitor to SR-BI, abolished the ability of RvT4 to promote macrophage cholesterol efflux. This was denoted by the inability of RvT4 to limit both intracellular lipid load and total cholesterol levels in the supernatants (Fig. 5g, h). Additionally, knockdown of SR-BI using an siRNA against its gene

*SCARB1*, blunted the ability of RvT4 to reduce macrophage-neutral lipid load and increase cholesterol efflux (Fig. 5i–k). Together these results highlight a role for SR-BI in mediating RvT4-initiated cholesterol efflux.

We next investigated the mechanism engaged by RvT4 to regulate SR-BI activity. Incubation of RvT4 with lipid-laden macrophages led to a rapid upregulation of SR-BI cell surface expression that reached a maximum within 15 min (Fig. 6a). To determine if this rapid response was reliant on remodeling of the actin cytoskeleton, we next incubated macrophages with Cytochalasin D prior to application of RvT4, to inhibit actin polymerization. Here we found that

**Fig. 5 | RvT4-mediated lipid efflux in oxLDL-loaded macrophages via SR-BI receptor.** Naïve macrophages obtained from peripheral blood monocytes. These were then either incubated with non-template controls (NTC) or siRNA targeting LDAH or nCEH. After 4 days **a** gene expression using real-time qPCR or **b** cells were incubated with DiI-labeled oxLDL for 16 h then incubated with vehicle or 1 nM RvT4 for 45 min. Macrophage lipid load, relating to the extent of efflux, was measured. **c** Macrophages were incubated with DiI-labeled oxLDL for 16 h and to evaluate the role of ABCA1 and ABCG1 we incubated the cells with Cyclosporine A, then with vehicle or 1 nM RvT4 for 45 min Macrophage lipid load evaluated. **c, d** Macrophages were incubated with siRNA to ABCA1, ABCG1 or NTC and after 4 days **d** the expression of ABCA1, ABCG1 was evaluated using real-time qPCR or **e, f** cells were incubated with DiI-labeled oxLDL for 16 h then incubated with vehicle or 1 nM RvT4 for 45 min and **e** cellular fluorescence and **f** total cholesterol in the supernatant were measured. **g, h** Macrophages were incubated with DiI-labeled oxLDL for 16 h and to evaluate the role SR-BI in cholesterol efflux cells were incubated with BLT-1 then with 1 nM RvT4 for 45 min, and **g** cellular fluorescence and **h** total cholesterol in the supernatant were measured. **i–k** Macrophages were incubated with siRNA to SR-B1 or NTC and after 4 days (**i**) the expression expression measured using flow cytometry. Representative histogram of SR-BI expression (white, siRNA control; gray, SCARB1 knockdown) or (**j, k**) cells were incubated with DiI-labeled oxLDL for 16 h then incubated with vehicle or 1 nM RvT4 for 45 min and (**j**) cellular fluorescence and (**k**) total cholesterol in the supernatant were measured. Error bars, mean ± s.e.m.; $n = 3$–5 per group, 2–3 independent experiments; Statistical differences were evaluated using repeated measures 2-way ANOVA and results were corrected for multiple comparisons by controlling the False Discovery Rate using the two-stage step-up method of Benjamini, Krieger and Yekutieli using non-normalized data. Statistical differences for data presented in panels b, c, e, g and j were determined using non-normalized values.

inhibition of actin remodeling limited the mobilization of SR-BI from intracellular pools to the plasma membrane (Fig. 6b). This decrease in plasma membrane expression of SR-BI was linked with a significant attenuation in the ability of RvT4 to reduce macrophage lipid load (Fig. 6c).

Cholesterol transported through macrophage SR-BI is unloaded onto HDL which then transports it to the liver[33]. Thus, we next tested the effect of HDL on RvT4-mediated cholesterol efflux. Co-incubation of lipid-loaded macrophages with HDL and RvT4 led to a decrease in neutral lipid content in human monocyte-derived macrophages when compared with cells incubated with RvT4 alone or HDL alone (Fig. 6d). By measuring HDL-associated cholesterol in the supernatants, we observed that RvT4 increased cholesterol loading onto HDL, an effect which was abolished in the presence of the SR-B1 inhibitor BLT-1 (Fig. 6e).

RA patients often present with metabolic dysfunction that is thought to exacerbate vascular disease[34]. Since WT mice are resistant to the development of atherosclerotic lesions, to test whether RvT4 regulates macrophage cholesterol efflux in vivo in settings relevant to the human RA we employed the *ApoE*[-/-] model that develops metabolic dysfunction and advanced atherosclerosis and combined this with K/BxN serum transfer arthritis. Administration of RvT4 to *ApoE*[-/-] arthritic mice fed WD led to an overall decrease in lipoprotein-associated cholesterol in plasma and an increase in the ratio of HDL-associated cholesterol to total cholesterol (Fig. 6f, g). HDL-associated cholesterol is subsequently excreted via the liver. Thus, we next evaluated whether RvT4 increased fecal levels of macrophage-derived cholesterol. To evaluate this, we transferred macrophages loaded with fluorescently labeled cholesterol into the peritoneum of *ApoE*[-/-] mice and quantified the fecal fluorescence levels after 72 h. Here we observed a marked increase in fecal-associated cholesterol in mice treated with RvT4 when compared with those that received vehicle alone, supporting the hypothesis that RvT4 promoted reverse cholesterol transport (Fig. 6h).

SR-BI plays a role in cholesterol uptake from HDL in the liver[35]. To explore whether the protective activities observed for RvT4 in vivo were also mediated via the regulation of hepatic SR-BI expression mice were injected with RvT4 and SR-BI expression in liver cells was measured using flow cytometry. Results from these experiments demonstrate that SR-B1 expression was not significantly regulated by RvT4 on hepatic cells (Fig. 6i).

Together these findings demonstrate that in macrophages RvT4 activates the translocation of SR-BI from intracellular pools to the plasma membrane, via the remolding of actin cytoskeleton, promoting the loading of intracellular cholesterol onto HDL, thus facilitating cholesterol efflux.

### Ablation of macrophage SR-B1 expression reverses the vascular protective activities of RvT4 in vivo

We next tested whether the vascular protective actions of RvT4 observed in WT mice fed a WD were retained in *ApoE*[-/-] mice, which display both advanced atherosclerotic lesions and increased joint disease during inflammatory arthritis[36]. We first evaluated whether RvT4 regulated key protective mechanisms in the resolution of vascular inflammation, namely efferocytosis, the reduction in macrophage lipid load, and the expression of macrophage activation markers. For this purpose, fluorescently stained apoptotic cells were injected into the peritoneal cavity of *ApoE*[-/-] mice, followed by vehicle or RvT4. Efferocytosis was determined after 45 min. Immunofluorescence analysis demonstrated a dose-dependent increase in the ability of RvT4 to upregulate macrophage efferocytosis of apoptotic cells in *ApoE*[-/-] mice (Supplementary Figs. 6 and 7a). Notably, this increase in macrophage efferocytosis was linked with a significant and dose-dependent increase in macrophage cell surface SR-BI expression (Supplementary Fig. 7b). Furthermore, overnight incubation of RvT4 with aortic sections from *ApoE*[-/-] fed WD led to a shift in macrophage phenotype with a downregulation of the activation marker CD11b, TGF-β1 and MHCII (Supplementary Fig. 7c).

Having observed that in vitro SR-BI expression was rapidly regulated by RvT4, a mechanism that was linked with the ability of this SPM to regulate cholesterol efflux in macrophages, we next questioned whether this mechanism was responsible for the vasculo-protective activities of RvT4 in vivo. For this purpose, we used a macrophage targeting siRNA approach to abrogate the expression of this receptor on macrophages in *ApoE*[-/-] mice fed a WD (see Fig. 7a). The effectiveness of the approach was corroborated using flow cytometry which highlighted a significant downregulation of SR-B1 expression in aortic macrophages from mice treated with siRNA targeting *Scarb1* when compared with mice treated with a control sequence (Fig. 7b). Quantification of vascular lipid load at week 16 after HFD initiation in arthritic *ApoE*[-/-] mice, demonstrated that administration RvT4 decreased both vascular lipid load and necrotic core area in mice that were also administered a CT siRNA sequence (Fig. 7c–e). Intriguingly, in mice receiving siRNA targeting macrophage *Scarb1* these protective actions of RvT4 were abrogated (Fig. 7c–e). Abrogation of SR-BI expression reversed the RvT4-elicit changes in HDL to total cholesterol ratio as well as plasma triglyceride and phospholipid concentrations (Fig. 7f–h). Taken together, these findings support a role for macrophage SR-BI in mediating the vasculo-protective properties of RvT4 during inflammatory arthritis.

### Discussion

Cardiovascular disease is one of the main co-morbidities in patients affected by rheumatoid arthritis[4] with the mechanisms contributing to this increased disease incidence remaining largely unexplained. In the present study, we report that RvT levels are negatively correlated with both the degree of joint and vascular disease. Administration of RvT4 to arthritic mice reduced vascular inflammation, an outcome that was linked with the regulation of macrophage lipid load and reactivity. Of note, pharmacological inhibition, or

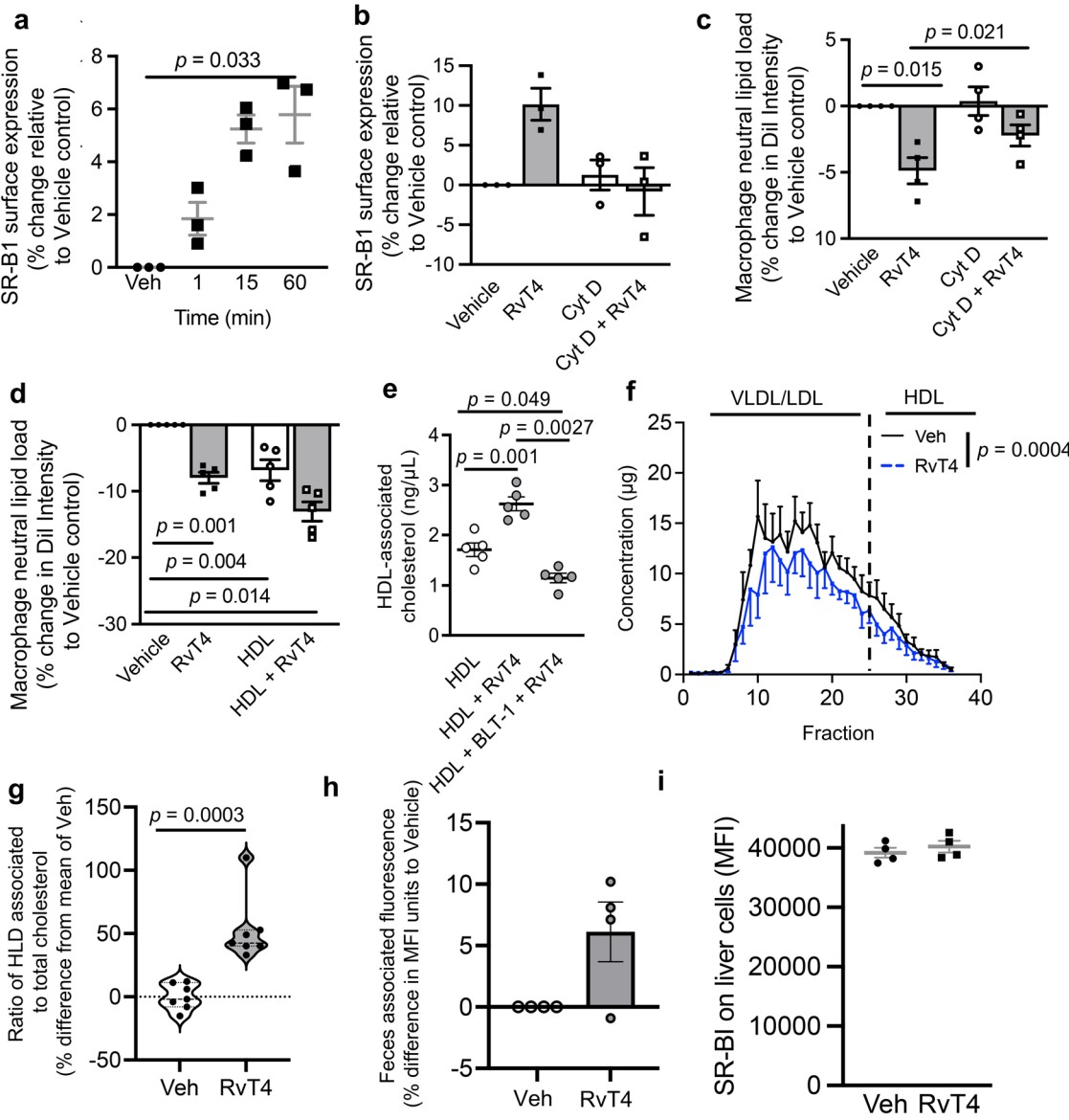

**Fig. 6 | RvT4 initiates actin-mediated SR-BI translocation to the cell surface.**
Monocyte-derived macrophages were incubated with DiI-labeled oxLDL for 16 h.
**a** Cells were incubated with vehicle or 1 nM RvT4 for 1, 15 or 60 min, and surface SR-BI expression measured using flow cytometry. $n = 3$ per group; Statistical differences were evaluated using 1-way ANOVA. DiI-labeled oxLDL-loaded macrophages were incubated with cytochalasin D and **b** SR-BI surface expression was measured, as well as **c** neutral lipid load. $n = 3-5$ per group; Statistical differences were evaluated using repeated measures 2-way ANOVA. To measure cholesterol efflux, DiI-labeled oxLDL-loaded macrophages were incubated with RvT4 and HDL and **d** cell-associated cholesterol measured. **e** Supernatants were collected and HDL-associated cholesterol was measured in the presence and absence of BLT-1. $n = 5$ per group; Statistical differences were evaluated using repeated measures 2-way ANOVA for (**d**) and 1-way ANOVA for (**e**). **f, g** *ApoE$^{-/-}$* mice were fed Western-style diet. K/BxN serum-induced arthritis was initiated and sustained for 4 weeks. After the final boost, mice were injected with vehicle or 75 ng RvT4 on alternating days for 10 days. Blood was collected and lipoproteins were separated using size exclusion chromatography fast protein liquid chromatographic. Cholesterol concentration in each fraction (y axis) was measured using a colorimetric assay and plotted against retention time (x axis). **f** Fractions corresponding to very-low-

density lipoprotein (VLDL), low-density lipoprotein (LDL), and high-density lipoprotein (HDL) are indicated. $n = 7$ per group; **g** The relative levels of cholesterol associated to HDL compared to total cholesterol were also quantified in these fractions. Statistical differences in (**f**) were evaluated using two-Way ANOVA and in (**g**) using unpaired *t*-test. **h** Raw 264.7 macrophages were loaded with fluorescently labeled cholesterol. These cells ($8 \times 10^6$ cells/mouse) were then transferred to *ApoE$^{-/-}$* mice via intraperitoneal injection. Mice were then treated with RvT4 (75 ng/mouse) or vehicle (PBS + 0.01% EtOH), feces were collected after 72 h and fluorescence evaluated using a plate-reader. $N = 4$ mice per group. Statistical differences were evaluated using Mann–Whitney test using non-normalized data. **i** Separately, *ApoE$^{-/-}$* mice were injected i.v. with 75 ng RvT4 or vehicle on days 0 and 2. Livers were collected on day 3 and single-cell suspensions prepared. Expression of SR-BI was measured in liver cells. $n = 4$ per group; Statistical differences were evaluated using unpaired *t*-test. Unless specified, error bars, mean ± s.e.m.; all ANOVA results were corrected for multiple comparisons by controlling the False Discovery Rate using the two-stage step-up method of Benjamini, Krieger and Yekutieli using non-normalized data. Results are from two distinct experiments. Statistical differences for data presented in panels b, c, d and h were determined using non-normalized values.

knockdown of macrophage SR-BI limited RvT4-initiated cholesterol efflux and prevented the protective actions of RvT4 both in vitro and in vivo. Together the findings elucidate a mechanism centered on the rapid activation of cholesterol efflux by RvT4 that leads to a

reprogramming of macrophage responses favoring the termination of vascular inflammation during inflammatory arthritis.

The role of macrophages in the onset and propagation of vascular and joint inflammation has been appreciated for some time[5,6,14,18]. While

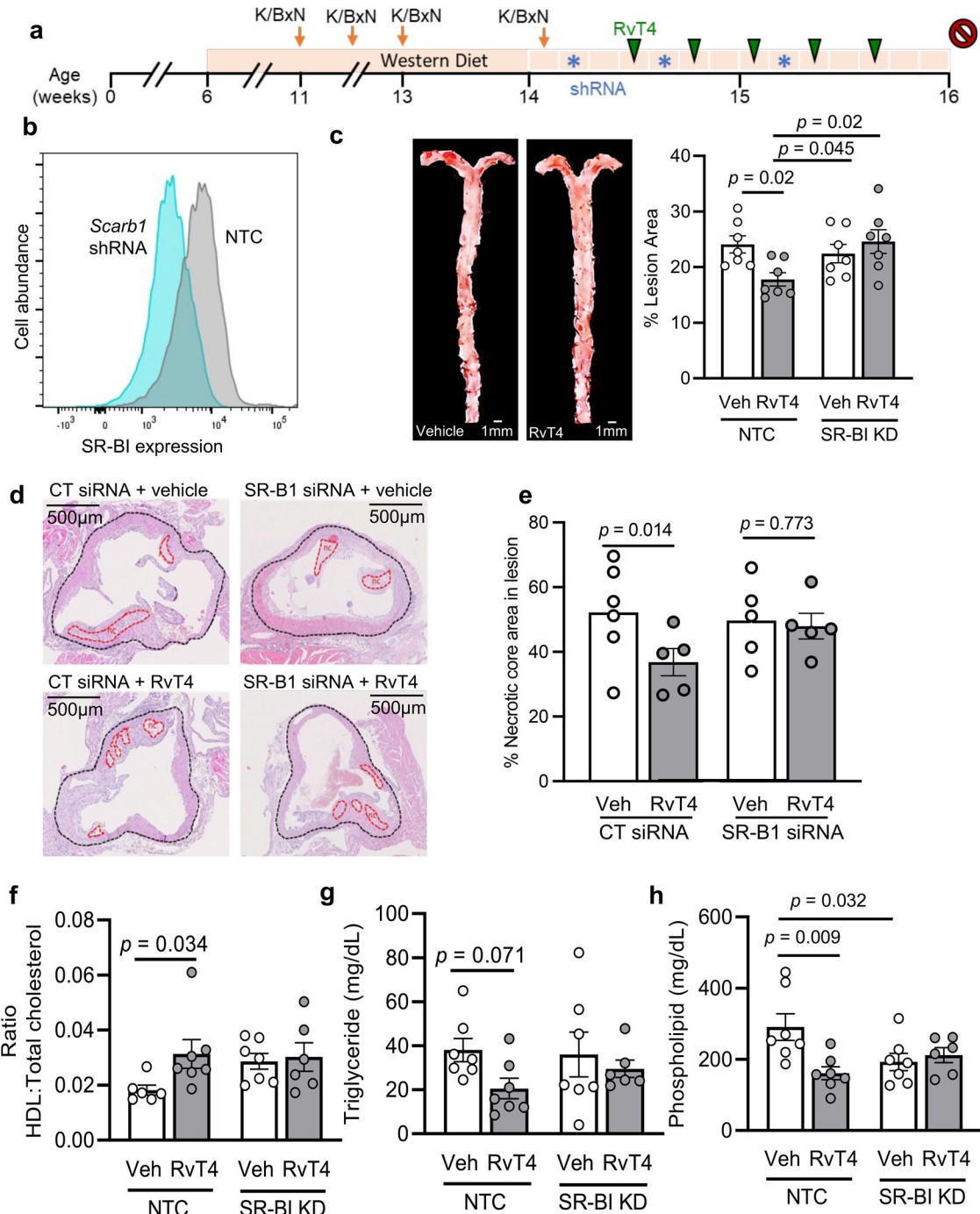

**Fig. 7 | RvT4 reduces lesion area and necrotic core in advanced atherosclerotic plaques.** *ApoE^-/-* mice were fed a Western-style diet, and K/BxN serum-induced arthritis was initiated and sustained for 4 weeks. Over a 12-day period after the final boost, mice were injected with in vivo-JetPEI-Man complexed with shRNA against *Scarb1* or a non-targeting control (NTC; blue asterisk), as well as 75 ng RvT4 or vehicle (green arrows). **a** Schematic representation of experiment. At the end of the experiment, **b** Aortas were collected, single-cell suspensions prepared, and expression of SR-BI on CD206-expressing cells was measured using flow cytometry. Representative histograms of SR-BI expression on CD206-expressing cells in the aorta 60 h after the first shRNA injection. **c** lesions were identified using oil-red O staining (left panel) representative images (right panel) lesion area was quantified

using ImageJ. *n* = 7 mice per group from two experiments. **d, e** Aortic roots were isolated, and sections obtained from -100 microns from the aortic sinus sections were stained with H&E, **d** representative sections necrotic core area were quantified and **e** lesion area was calculated using ImageJ. *n* = 5–6 per group from two experiments. Statistical differences were evaluated using 2-way ANOVA. **f–h** Plasma was collected and **f** cholesterol, **g** triglyceride, and **h** phospholipid quantified in control and SR-BI knockdown mice. *n* = 7 per group Results are from two experiments. Statistical differences were evaluated using 2-way ANOVA corrected for multiple comparisons by controlling the False Discovery Rate using the two-stage step-up method of Benjamini, Krieger and Yekutieli. Error bars are mean ± s.e.m.

the ontogeny of these cells in human tissues is still subject of investigation, in mouse diseased tissue macrophages arise from two main origins. The first is embryonic, and this subset of macrophages is referred to as tissue resident macrophages. The second population is

derived from circulating monocytes that infiltrate the inflamed tissues and differentiate to macrophages. While tissue resident cells play an important part in the initial stages of inflammation, it is monocyte-derived macrophages that are thought to play a central role in

the propagation and maintenance of tissue inflammation. In early atherosclerosis, apoB-containing lipoproteins are retained in the subendothelial matrix of medium-to-large arteries and stimulate a robust inflammatory response, resulting in blood-borne leukocyte recruitment[37]. As atherogenesis progresses, this local inflammatory circuit response becomes self-perpetuating, further sustaining monocyte recruitment. Despite the continued recruitment of monocytes and their differentiation into macrophages, the inflammatory environment in atherosclerotic plaques and that found in inflamed joints, influences cellular polarization. This aspect is typified by an impairment of macrophage protective functions in these tissues, including their ability to uptake and clear apoptotic cells[5,6,14,18].

In addition to being central in the onset and perpetuation of vascular inflammation, recent studies demonstrate that macrophages also contribute to the termination of inflammation[38–40]. In the present study, we demonstrate that administration of RvT4 to arthritic mice leads to a reduction in both joint and vascular disease. This improvement in disease activity was linked with a shift in macrophage phenotype. In vascular tissue-derived macrophages, RvT4 upregulated expression of the phagocytic receptor, MerTK, which is involved in the resolution of vascular disease by facilitating the uptake and clearance of apoptotic cells[25]. In these cells, we also observed a significant downregulation in the expression of CD11c. Notably, recent evidence demonstrates that upregulation of CD11c in macrophages is linked with the propagation of vascular inflammation during atherosclerosis[27]. Together, our findings underscore a role for RvT4 in promoting macrophage phenotype switch in vascular tissues to facilitate the termination of locally propagated inflammatory processes.

Macrophage uptake of lipoproteins, in particular oxidized lipoproteins, leads to accumulation of cytoplasmic neutral lipid droplets, making the cells appear 'foamy', a typical histological hallmark of atherogenesis. Intracellular lipid accumulation in macrophages can further dysregulate the ability of macrophages to uptake and clear apoptotic cells while concomitantly upregulating their pro-inflammatory profile[5,6,14,18]. In addition, lipid-laden macrophages are unable to traffic out of the inflamed tissues, leading to cell accumulation and perpetuation of local inflammation. These foamy macrophages will ultimately become apoptotic and given that atherosclerotic tissue macrophages display impaired efferocytosis, lack of or delayed clearance of these apoptotic cells will lead to cell necrosis, further aggravating the local inflammatory milieu.

Cholesterol removal from atherosclerotic plaques can occur via several routes. The first is via efflux of cholesterol from aortic macrophages and loading onto HDL particles. As atherogenesis progresses, this mechanism becomes disrupted. A second mechanism relies on the trafficking of cholesterol-loaded macrophages out of the vascular tissues, an aspect that also becomes disrupted with disease progression, since lipid-laden macrophages become progressively impaired in their ability to respond to chemotactic queues as lipid load increases[5,6,14,18]. Here we report that RvT4 rapidly increased the hydrolysis of neutral lipids to release esterified cholesterol via the upregulation of nCEH activity. Since intracellular free cholesterol is toxic to cells, its concentrations are tightly controlled. One mechanism that plays a central role in this process is cholesterol efflux. Intracellular free cholesterol is actively transported out of cells and loaded onto acceptor HDL particles which are then taken up in the liver[29,30]. In isolated macrophages, as well as tissue macrophages obtained from atherosclerotic aortic tissues, RvT4 increased cholesterol efflux via the regulation of SR-BI externalization. Indeed, inhibition of this receptor, using either a pharmacological or siRNA approach, reversed the ability of RvT4 to reduce macrophage-neutral lipid load in both isolated and tissue macrophages. This mechanism was linked with a reprogramming of macrophages towards a tissue protective phenotype, increasing their ability to migrate towards chemotactic signals and to uptake apoptotic cells. These changes in

macrophage responses were also linked with enhanced tissue protection as demonstrated by decreased lesion area and necrotic core area in atherosclerotic plaques in vivo.

For many years the role of SR-BI in regulating cholesterol homeostasis and protecting against vascular inflammation was primarily linked with uptake of cholesterol esters in the liver. Recent findings have challenged this hypothesis indicating that, in the context of atherosclerosis, SR-BI mediates cholesterol efflux from macrophages[41]. Deletion of *Scarb1*, the gene encoding SR-B1, in macrophages results in defective efferocytosis and increased plaque size, necrotic core area and vascular inflammation[42]. This extra hepatic role for SR-BI is further supported by findings made in humans whereby patients with a functional mutation in this protein, that leads to virtually complete loss of function of SR-B1, are at an increased risk of cardiovascular disease despite elevated HDL-associated cholesterol levels[33]. Such observations have been confirmed in studies conducted with global SR-BI knockout mice, whereby these animals display an increase in cardiovascular disease despite elevated plasma levels of HDL-associated cholesterol[35]. Endothelial SR-BI mediates the transport of both HDL and LDL to the subintimal space to modulate the extent of atherosclerosis[43]. Our findings lend support to this mounting body of evidence, demonstrating that through regulation of SR-B1, RvT4 reprograms macrophage responses towards a pro-resolving and tissue protective phenotype. In vivo, this results in the protective effects of RvT4 on aortic lesions and plasma lipids, which were abolished when SR-BI expression was knocked down.

In summation, we present here compelling evidence that RvT4 rapidly upregulates both neutral lipid breakdown, via the activation of nCEH, and the removal of free cholesterol from macrophages by promoting the translocation of SR-BI to the plasma membrane. This mechanism results in functional reprogramming of macrophages favoring their pro-resolving activities. Such changes are linked with a decrease in vascular disease, as evidenced by a reduction in vascular lesions and decrease in necrotic core size in the aortas of mice treated with RvT4. Collectively, the present findings elucidate a protective mechanism centered around the RvT4-SR-BI axis, that reprograms lipid-laden macrophages to limit morbidity in inflammatory arthritis.

## Methods
### Ethics
Research performed in these studies complies with relevant ethical regulations. Research performed with animals was conducted in accordance with protocols detailed in a UK Home Office approved protocol (P998AB295). Leukocyte cones were obtained from consented volunteers from the National Health Service Blood and Transplant, St. George's Hospital, London, to prepare human monocyte-derived macrophages. Experiments with human cells were performed in accordance with protocols approved by Queen Mary Ethics of Research Committee (QMERC2018/70; QMERC2020/49; QMERC22.331).

### Animals
C57BL/6 mice and were purchased from Charles River Laboratories (Kent, United Kingdom) and B6.129P2-*Apoe*^tm1Unc^/J (ApoE^-/-^) from Jackson Laboratories (JAX stock Cat no: 002052). Experiments strictly adhered to United Kingdom Home Office regulations (Guidance on the Operation of Animals, Scientific Procedures Act), Laboratory Animal Science Association Guidelines (Guiding Principles on Good Practice for Animal Welfare and Ethical Review Bodies) and according to protocols detailed in a UK Home Office approved protocol (P998AB295).

All animals were provided with water ad libitum and either standard laboratory diet or Western-style diet (Special Diets Services, Cat no: 829100) from 6 weeks age. Mice were kept on a 12-h light/dark cycle. Mice were euthanized using $CO_2$ followed by cervical dislocation.

## Inflammatory arthritis

K/BxN serum was produced by breeding NOD/Shiltj mice with KRN transgenic mice as in ref. 44. Serum from subsequent K/BxN offspring was collected at 10 weeks age and used to initiate inflammatory arthritis in recipient mice.

Disease progression was measured daily using a 26-point clinical scoring system, counting the total number of swollen ankles/wrists, pads, and digits.

To initiate arthritis, recipient 11-week old male C57BL/6 or ApoE$^{-/-}$ mice, fed either standard laboratory chow or WD for 5 weeks prior to initiation, were administered K/BxN serum (100 μl, i.p.) on days 0 and 2. In select experiments, arthritis was sustained by giving and additional three weekly injections of arthritogenic serum. After the final injection, mice were injected i.v. with vehicle (DPBS$^{-/-}$) or RvT4 (75 ng) on alternating days for 10 days. At the experimental endpoint, mice were anesthetized using 3% isoflurane and blood collected via cardiac puncture into heparin-lined syringes. Mice were then culled, and tissue collected.

## Histology

Tissues (heart, aorta and joints) were placed in 10% neutral buffered formalin (v/v; in water that contained 0.65% Na$_2$HPO$_4$ and 0.4% NaH$_2$PO$_4$) for 48 h. Fat surrounding the aorta was removed. The heart was excised and embedded in paraffin wax. Sections were obtained 100 microns from the aortic sinus.

Bony tissues were decalcified by placing in 10% EDTA in phosphate-buffered saline (w/v) for 2 weeks, and solution replaced every 3 days until bones were decalcified. Tissues were then embedded in paraffin wax and sectioned to 4-micron thickness.

Safranin O stain of aortas was carried out as previously described[45]. Briefly, deparaffinized and dehydrated sections were incubated in 0.1% Safranin O (w/v; in 164 mM acetic acid and 36 mM sodium acetate, pH 4.0) for 5 min. Sections were then counter-stained with 0.05% Light Green stain for 3 min, and coverslips mounted. Images were captured using NanoZoomer S210 (Hamamatsu) and Safranin O intensity quantified using ImageJ software.

Hematoxylin and eosin staining was carried out by the Barts Cancer Institute Pathology Core as previously described[16]. Images were captured using NanoZoomer S210 (Hamamatsu) and necrotic area quantified using ImageJ software.

Aortic arches were dissected longitudinally and incubated with 0.3% oil-Red O (w/v; in 60% propan-2-ol containing 0.6% dextrin) for 20 min. Excess oil-Red O was removed and *en face* images captured using bright field microscopy. Lesion area and staining intensity was quantified using ImageJ software.

## Lipid mediator profiling

Lipid mediators were extracted, and profiling conducted as previously described[46]. Briefly, ice-cold methanol containing 500 pg of each deuterated (d) internal standard was added to samples: d$_8$–5S-hydroxyeicosatetraenoic, d$_4$-LTB$_4$, d$_5$-LXA$_4$, d$_4$-PGE$_2$, and d$_5$-RvD2. These were used to facilitate mediator identification and quantification as they represent the chromatographic regions of interest. Precipitated proteins were removed by centrifugation, and supernatant subjected to C-18 solid-phase extraction techniques using ExtraHera (Biotage). Lipid mediators were identified and quantified using liquid chromatography-coupled tandem mass spectrometry. Multiple reaction monitoring was carried out by using signature Q1 (parent ion) and Q3 (characteristic daughter ion) ion pairs for each molecule, acquired in negative ionization mode. In Sciex OS-Q, raw data files were processed using the 'noise filtering' and 'low' smoothing functions. Peak picking was performed using the 'AutoPeak' algorithm and then manually inspected. Where the peak identified by the 'AutoPeak' algorithm was not that corresponding to the mediator of interest, the correct peak was integrated manually. Lipid mediators were identified

based on matching retention time of the peak of interest with those of authentic or synthetic standards and a signal-to-noise ratio ≥5. Signal-to-noise ratios were calculated using the 'Relative Noise' algorithm. The robustness of this methodology was evaluated against the Peak-to-Peak method and found to give essentially similar findings when determining which molecules are below the ≥5 signal-to-noise ratio cutoff (Supplementary Fig. 10). Furthermore, this relative noise algorithm overcomes some of the limitations displayed by the Peak-to-Peak method related to the identification of an appropriate region within the chromatogram to define the 'noise'. In samples where there are several closely eluting biological isomers, as is the case for many of the lipid mediators, identifying a region close to the peak of interest that is of sufficient width to accurately calculate the noise component of the chromatogram can be challenging. Furthermore, the use of the relative noise algorithm also overcomes the subjectivity in selecting the noise region and utilizes the entire chromatogram to identify the noise under the peak of interest.

Quantitation was carried out in accordance with published methods that included calculating recoveries of deuterium-labeled internal standards and linear calibration curves. Where curves for mediator of interest were not available calibration curves for surrogate molecules with similar physical characteristics were employed. Namely for the quantitation of PDx and 17R-PD1 we used standard curves obtained for PD1, for 4S, 14S-diHDHA we employed the standard curve for MaR1, for 17R-RvD1 that for RvD1, for 15-epi-LXA$_4$ that for LXA$_4$ and for 15-epi-LXB$_4$ that for LXB$_4$. Supplementary Table 4 reports the deuterium labels that were used to determine the recoveries of each of the mediators evaluated in these studies.

## Biogenic RvT4 production

13R-HDPA methyl-ester (Kindly provided by Prof. Trond Vidar Hansen, University of Oslo, Norway), was saponified using 1 M lithium hydroxide. The resulting 13R-HDPA was isolated using reversed-phase high-performance liquid chromatography (RP-HPLC), utilizing a 1260 Infinity LC System (Agilent Technologies) and Agilent C-18 Poroshell column (2.7 μm × 4.6 mm × 150 mm) with the column oven set to 50 °C and a chromatographic gradient as described in as in ref. 15.

Solvents were removed using a gentle stream of nitrogen, and isolated 13R-HDPA suspended in phosphate buffer (KH$_2$PO$_4$, 0.1 M, pH 6.3)[15] and incubated with potato 5-lipoxygenase for 40 min, in ice water. The incubation was stopped using 2 volumes of ice-cold methanol and precipitate removed by centrifugation for 10 min, 4000 × g, 4 °C. The supernatant was then subjected to solid-phase extraction and RvT4 isolated using RP-HPLC, with elution monitored using a UV spectrophotometer as in ref. 15.

To characterize the material isolated we used a signal-to-noise ratio ≥5 for the peak obtained in liquid chromatography that matched the retention time of reference material as detailed above. To further validate the biogenic material used we matched the MS/MS spectrum obtained for the product under the peak with that from reference material using Sciex OS library function and a cutoff value for the library match score ≥70%. We also matched the UV chromophore obtained in static UV to that of reference material[15] (Supplementary Fig. 3).

## Ex vivo tissue incubations

C57BL/6 or ApoE$^{-/-}$ mice were kept on Western Diet for 7 weeks to induce vascular inflammation. In select experiments, K/BxN serum transfer-induced arthritis was initiated and mice culled days 6 after initiation. Aortas were excised and surrounding adipose tissue carefully removed. Aortas were then placed in RPMI-1640 containing 1% penicillin, 1% streptomycin, 1% amphotericin B, Modified Eagle's Media with non-essential amino acids, and 2 mM L-glutamine and incubated with vehicle (DPBS$^{-/-}$) or RvT4 (1 nM). After 30 min (at 37 °C, 5% CO$_2$). FBS was then added to a final concentration of 10% and tissues

incubated for a further 18 h. At the end of the incubations, cells that had migrated out of the tissue onto the well were collected and enumerated. We also liberated cells from the tissues and cells were then stained to identify the macrophage population using the protocols detailed below. In select experiments macrophages were pretreated with vehicle (DPBS$^{-/-}$) or BLT-1 (0.2 μM) for 30 min prior to RvT4 treatment.

## Preparation of single-cell suspensions

Aortic tissues from ex vivo experiments were cut to approximately 2 mm lengths and placed in a digestion buffer of RPMI-1640 containing 1.25 mg/mL collagenase D and 0.2 mg/mL DNase I. Tissues were incubated at 37 °C for 75 min in a horizontal shaker set to 250 rpm, and then passed through a 70 μM strainer and cells collected in RPMI-1640 containing 10% fetal bovine serum and centrifuged at 500 × g for 10 min.

Livers were isolated and perfused with calcium-free EDTA perfusion medium (0.05% KCl, 10 mM HEPES buffer, 5 mM D-glucose, 200 μM EDTA, in sterile DPBS without calcium or magnesium) via the portal vein. Collagenase-containing perfusion medium was then run through (perfusion medium above containing 0.33 mg/mL Collagenase II). Liver cells were dissociated using fine forceps and cells run through a 70 μm filter. Cells were washed by centrifugation three times, and cell count and viability determined.

## Immuno-fluorescent staining for Flow cytometry

Isolated cells were suspended in DPBS$^{-/-}$ containing 0.02% bovine serum albumin and 1% Fc-blocking IgG (v/v) and incubated with 0.1% LIVE/DEAD Fixable Stain for 20 min on ice. Excess stain was removed, and cells incubated with fluorescently labeled antibodies for 30 min on ice. Cells were washed and then fixed using 1% paraformaldehyde. In select experiments, LipidTOX Green Neutral Lipid Stain (Thermo-Fisher) was added to cells at 1× and incubated for 30 min. CountBright Absolute Counting Beads were used for leukocyte enumeration. Staining was evaluated using LSRFortessa cell analyzer and analyzed using FlowJo software.

## Gene expression

Samples were collected in tubes containing Lysing Matrix E (MP Biomedicals) and homogenized using a BeadBeater. RNA was extracted using RNeasy Mini Kit (Qiagen; Cat no: 74104) and cDNA generated using SuperScript III Reverse Transcriptase (Invitrogen). Gene expression was assessed using quantitative real-time PCR (qRT-PCR) analysis with SYBR green I fluorescent dye, and QuantiTect Primer Assays (Qiagen) for the following mouse genes: *Mertk, Adam17, Thbs1, Mfge8*, and *Icam1* and human *Abca1, Abcg1, Rplp13, c2orf43 (LDAH) and nCEH*. The qRT-PCR reaction was carried out using a 7900HT Fast Real-Time PCR System (Applied Biosystems). Target gene expression was normalized to *Gapdh* for mouse experiments and *Rplp13* for human experiments and expressed as a relative value.

## Macrophage preparation

Mouse bone marrow-derived macrophages were prepared by collecting bone marrow cells from C57BL/6 or ApoE$^{-/-}$ mice and seeding in sterile dishes. Cells were incubated in a humidified chamber in 5% CO$_2$ at 37 °C for 2 h, and non-adherent cells removed. RPMI-1640 containing 10% FBS, 1000 U penicillin, 100 mg/mL streptomycin and 20 ng/mL recombinant mouse GM-CSF (PeproTech, Cat no: 315-03) was added, and media replaced on day 4.

Leukocyte cones were obtained from consented volunteers from the National Health Service Blood and Transplant, St. George's Hospital, London, to prepare human monocyte-derived macrophages. Experiments with human cells were performed in accordance with protocols approved by Queen Mary Ethics of Research Committee

(QMERC2018/70; QMERC2020/49; QMERC22.331). PBMC were isolated by centrifugation for 30 min at 400 × g, 21 °C, using Histopaque 1077 (Sigma–Aldrich), and seeded in sterile dishes. Cells were differentiated for 7 days in RPMI-1640 containing 10% human serum, 1000 U penicillin, 100 mg/mL streptomycin, and 20 ng/mL human recombinant GM-CSF (PeproTech, Cat no: 300-03), in a humidified chamber in 5% CO$_2$ at 37 °C, replacing media on day 4.

## siRNA-mediated gene knockdown

Accell Human siRNA SMARTpool (Horizon Discovery) against *Scavenger Receptor Class B Member 1 (SCARB1*; Cat no: E-010592-00-0010), *Lipid-Droplet-Associated Hydrolase (LDAH*; Cat no: E-009926-00-0005), or *Neutral Cholesteryl Ester Hydrolase (nCEH*; Cat no: E-005245-00-0005) *ATP-binding Cassette A1 (ABCA1*; Cat no: E-004128-00-0010), *ATP-binding Cassette G1 (ABCG1*; Cat no: E-008615-00-0010), or a non-targeting control were used to knockdown protein expression in accordance with manufacturer's instructions. Briefly, Accell Delivery Media containing 1 μM SMARTpool siRNA was incubated with cells seeded at 1500 cells/mm$^2$ for 4 days. Protein knockdown was confirmed using flow cytometry and/or functional assays.

## In vivo-JetPEI-Man mediated gene knockdown

Plasmids containing 5 validated TRC clones of MISSION shRNA (Sigma–Aldrich) against mouse *Scarb1* (Cat no: TRCN0000066574, TRCN0000066575, TRCN0000066576, TRCN0000066577), as well as MISSION TRC2 pLKO.5-puro Empty Vector Control Plasmid DNA (Cat no SHC201), were expanded. Plasmids were isolated using ZymoPURE II Plasmid Midiprep Kit (Cat no: D4200). For each animal, 30 μg nucleic acid was delivered intravenously in complex with in vivo-JetPEI-Man reagent prepared according to manufacturer's instructions, at N/P ratio of 6. Mice were monitored for potential toxicity.

## Cholesterol efflux assay

These experiments utilized naïve macrophages or macrophages deficient in *nCEH, LDAH, SR-BI, ABCA1, or ABCG1*. On day 6 of macrophage differentiation with GM-CSF, lipid-loaded cells seeded at 1500 cells/mm$^2$ were prepared by adding 10 μg/mL DiI-labeled oxLDL (Invitrogen) for 16 h. Excess oxLDL was then removed and media replaced. In select experiments, macrophages were incubated with Cyclosporine A (8 μM), BLT-1 (0.2 μM), or Cytochalasin D (25 nM) for 30 min, and then with vehicle, HDL (25 μg/mL), or RvT4 (1 nM) for 45 min, 5% CO$_2$, 37 °C. In select experiments, cell supernatants were collected for cholesterol quantification. Media was replaced with fresh RPMI and remaining cellular fluorescence was measured using a NOVOstar plate-reader. In other experiments, macrophages were prepared and loaded with DiI-labeled oxLDL as detailed above and incubated with vehicle, RvT4 (1 nM) or PGE$_2$ (1 nM) for 45 min, 5% CO$_2$, 37 °C prior to determination of intracellular cholesterol as detailed above.

## In vivo cholesterol efflux

ApoE$^{-/-}$ mice fed WD were injected intraperitoneally with 5 μg CholEsteryl BODIPY-FL C12 (ThermoFisher Scientific, Cat no: C3927MP) in 500 μL DPBS$^{-/-}$. After 16 h, mice were injected i.p. with vehicle or 75 ng RvT4 in 100 μL DPBS$^{-/-}$. After 2 h, peritoneal lavages were carried out, and cells separated by centrifugation.

In separate experiments, RAW 264.7 were incubated with 5 μg/mL 22-NDB Cholesterol in complete RPMI-1640 for 16 h (37 °C). Cells (8 × 10$^6$ cells/mouse) were then transferred to ApoE$^{-/-}$ mice via intraperitoneal injection. Mice were treated with either vehicle or RvT4 (75 ng/mouse) via intraperitoneal injection and feces were collected after 72 h. Samples were then homogenized in NDB-Cholesterol buffer (V:V:V; 80:19:16 isopropanol:hexane:125 mM H$_2$SO$_4$) as previously described[47] and fluorescence was evaluated using an excitation wavelength of 470 nm and an emission wavelength of 530 nm.

## Macrophage efferocytosis

Apoptosis was induced in HL-60 cells (ATCC, Cat no: CCL-240) by exposure to UV-C light (254 nm) for 15 min, then incubated in RPMI-1640 containing 10% FBS for 2 h at 5% $CO_2$, 37 °C as previously reported[48].

For in vitro experiments, apoptotic cells were labeled using pHrodo Red, succinimidyl ester (ThermoFisher, Cat no: P36600) for 30 min. oxLDL-loaded macrophages, seeded at 470 cells/mm², were labeled with CellBrite Green (Biotium, Cat no: 30021) at 200× dilution for 60 min, 5% $CO_2$, 37 °C, and excess stain removed. Cells were then incubated with vehicle (DPBS$^{-/-}$) or RvT4 (0.1, 1 or 10 nM) for 10 min, and apoptotic cells added at a 3:1 ratio of apoptotic cells to macrophage count. Efferocytosis was monitored using Zeiss Celldiscoverer 7, over 30 min.

For in vivo efferocytosis, apoptotic cells were labeled with PKH26 Red Fluorescent Cell Linker Kit (Sigma–Aldrich, Cat no: PKH26GL) as per manufacturer's instructions. Briefly, apoptotic cells were suspended in Diluent C containing PKH26 dye and mixed gently for 4 min. Staining was stopped by adding 1% bovine serum albumin. ApoE$^{-/-}$ mice were injected intraperitoneally with either vehicle (DPBS$^{+/+}$) or RvT4 (75 or 150 ng) and left for 5 min, and then with 6 million apoptotic cells. After 1 h the peritoneum was lavaged using DPBS$^{-/-}$, cells collected and efferocytosis, SR-BI and lipid load in peritoneal macrophages were evaluated using flow cytometry as above.

## Chemotaxis

Mouse macrophages were incubated with vehicle (DPBS without calcium and magnesium) or RvT4 (1 and 10 nM) for 30 min, 37 °C, prior to addition of oxLDL. A ChemoTx Disposable Chemotaxis System plate (Neuroprobe) with 5-µm pores was used to evaluate chemotaxis. Briefly, 29 µL of assay medium containing vehicle (DPBS), MCP-1 (25 ng/mL) or ATP (50 µM) as chemoattractants was transferred into selected wells. The 5-µm pore membrane was placed and 15 µL cell suspension containing ~30,000 cells was gently pipetted into the lumen of each chamber. The lid was placed, and the plate incubated for 6 h, 5% $CO_2$, 37 °C. The number of transmigrated cells was quantified using PrestoBlue following manufacturer's instructions and a NOVOstar plate-reader.

## Cholesterol quantification

Intracellular and extracellular cholesterol or cholesteryl ester concentrations were quantified using a fluorometric Cholesterol/Cholesteryl Ester Assay Kit (Abcam, Cat no: ab65359).

In select experiments, lipids were extracted from 100,000 oxLDL-loaded cells using chloroform:isopropanol:Igepal at a ratio of 7:11:0.1. The extract was centrifuged for 10 min at 15,000 × g, and the organic phase collected. Solvent was removed using a gentle stream of nitrogen, and lipids suspended with assay buffer.

Reaction mixes to quantify total cholesterol (containing cholesterol esterase) and free cholesterol (without cholesterol esterase) were prepared in accordance with manufacturer's instructions, added to samples, then incubated for 60 min at 37 °C, in the dark. Fluorescence was measured using a NOVOstar plate-reader, and cholesterol concentrations calculated using a standard curve.

Lipoproteins were separated from plasma using size exclusion chromatography at 4 °C. An AKTA pure fast protein liquid chromatography system (GE Healthcare) equipped with a Superose 6 Increase 3.2/300 column (GE Healthcare, Cat no: 29-0915-98) was used with a running buffer of 10 mM Tris-HCl, 150 mM NaCl, pH 7.4. Chromatography was carried out using filtered plasma, with a flow rate of 0.4 mL/min under a pressure of 150 psi[49]. Fractions of 50 µL were collected and the concentration of total cholesterol in the eluted fractions measured using Total Cholesterol Reagents (ThermoFisher Scientific, Cat no: TRI3421) or Cholesterol/Cholesterol Ester-Glo Assay (Cat no: J3190) following manufacturers' instructions for the measurement of total cholesterol concentrations.

In experiments presented in Fig. 7 HDL and LDL/VLDL associated cholesterol were determined using the HDL and LDL/VLDL Cholesterol Assay Kit (Abcam, Cat no: ab65390) following manufacturer's instruction.

## Plasma lipid quantification

Plasma triglycerides were measured using Triglyceride Assay Kit (Abcam, Cat no: ab65336) according to manufacturer's instructions. Phospholipids were measured using Phospholipid Quantification Assay Kit (Sigma–Aldrich, Cat no: CS0001), according to manufacturer's instructions. Assay fluorescence and absorbance was measured on a NOVOstar plate-reader, and concentrations calculated using a standard curve.

## Statistics

Results are presented as mean ± s.e.m. Statistical analyses were carried out using GraphPad Prism 9 Software. Differences between two groups were tested using unpaired *t*-test or one-sample *t*-test. If more than two groups, statistical significance was assessed using 1-way or 2-way ANOVA (if 2 nominal variables were tested). All ANOVA results were corrected for multiple comparisons by controlling the False Discovery Rate using the two-stage step-up method of Benjamini, Krieger and Yekutieli. Linear regressions were used to calculate correlations. The number of subjects used in each experiment was determined by carrying out power calculations using historical data presenting strong effect sizes, to reach a power of 80%. Principal Component Analyses (PCA) were carried out using MetaboAnalyst[50]. Within the data matrices, compounds quantified using LC-MS/MS were normalized by weight or plasma volume. If compounds were below the lower limits of quantitation (i.e., s/n > 5), they were denoted as zero in the statistical tests. For PCA analysis mediators that were identified in less than 50% of the samples were excluded. SPMs that were below limits of detection were replaced by 1/5 of lowest values, data was normalized by the mean for each parameter and then scaled using auto scaling (mean-centered and divided by standard deviation of each variable) in MetaboAnalyst. For PCA of expression markers, data was normalized by sum and scaled using the AutoScaling feature in MetaboAnalyst.

## Reporting summary

Further information on research design is available in the Nature Portfolio Reporting Summary linked to this article.

# Data availability

The data generated in this study are provided in the article and its supplementary files or from the corresponding author upon request. Raw mass spectrometry files are available from EBI BioStudies Accession number: S-BSST880. Source data are provided with this paper.

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

## Acknowledgements

This work was supported by funding from the European Research Council (ERC) under the European Union's Horizon 2020 research and innovation programme (grant no: 677542) and the Barts Charity (grant no: MGU0343 and MGU0439). J.D. is also supported by a Sir Henry Dale Fellowship jointly funded by the Wellcome Trust and the Royal Society (grant 107613/Z/15/Z). This work acknowledges the support of the National Institute for Health Research Barts Biomedical Research Centre (NIHR203330).

## Author contributions

J.D. designed the experiments and conceived the overall research plan; M.E.W. and R.D.M. conducted the experiments; M.E.W., R.D.M, M.P and J.D. analyzed results; all authors contributed to manuscript preparation.

## Competing interests

J.D. is an inventor on a patent application (WO/2017/015271) related to the composition of 13-series resolvins and their use in treating uncontrolled inflammation that is assigned to Brigham and Women's Hospital Boston USA. J.D. and M.W. are inventors on a patent application (WO/2018/197650) detailing the diagnostic utility of 13-series resolvins as biomarkers. Other authors declare no competing interests.
