## [Peer Review File · Nature Communications]

Resolvin T4 enhances macrophage cholesterol efflux to reduce vascular diseaseREVIEWER COMMENTS

Reviewer #1 (Remarks to the Author):

This study investigated the effect of RvT4 on macrophage lipid accumulation, joint inflammation and aortic atherosclerosis. The authors report that RvT4 may promote cholesterol efflux in foam cells via SR-BI, thereby reprogramming lipid laden macrophages and yielding tissue protection. Although I find the study novel and interesting and of potential wide interest, some major conclusions are not supported by the data, and the study of lipoproteins (including HDL, cholesterol efflux and reverse cholesterol transport) and atherosclerosis are in need of a major improvement.

Major points:

1. The effects of RvT4 in vivo on plasma lipids and lipoproteins should be shown (free and total cholesterol, triglycerides, and cholesterol in plasma lipoprotein fractions) in mice studies.
2. The idea that RvT4 is affecting macrophages SR-BI only in articulations and vasculature needs to be proven. More specifically, authors should rule out that RvT4 is also affecting liver SR-BI, where it would have the potential to induce major changes in circulating HDL levels.
3. Figure 5 data would seem critical for concluding that RvT4 is promoting macrophage cholesterol efflux via SR-BI. However, the authors did not determine cholesterol efflux. Only in panel A, the authors demonstrated that RvT4 promoted cholesterol release into the medium, but the rest of panels are determining macrophage neutral lipid accumulation, not the ability of these macrophages to promote macrophage cholesterol efflux. The legend is difficult to follow. For example, in panel I the authors indicated that macrophages were incubated with ox-LDL then with 1nM RvT4 and lipid efflux was assessed; but lipid efflux is not shown (only neutral lipid load). Which are the acceptors used in the different experiments: whole plasma, isolated HDL?

There is a need of studying cholesterol efflux directly by using standard methodology. There are ways to distinguish the efflux ABCA1 or ABCG1 dependent that will need to be performed.

4. I do not think that macrophage SR-BI is currently considered a major mechanism for cholesterol efflux. In fact it is usually included on the term "unspecific efflux". Since ABCA1 and ABCG1 are in contrast critical in specific cholesterol efflux, they should be directly determined both in vitro and in vivo.

5. Cholesterol efflux is only the first RCT step. Therefore, the authors need to show in mice that RvT4 promotes increase in macrophage-specific reverse cholesterol transport. This would be the physiological mechanism that could explain, at least in part, atherosclerosis improvement.

6. In the manuscript, atherosclerosis area is quite more extensive than usual under Western diet. Wild-type mice usually develops early fatty streak under a Western-type diet and this is mainly located in proximal aorta. ORO staining intensity is not a relevant measure of atherosclerosis extension (rather the % of area covered by lesions).

7. In figure 7, the authors used hyperlipidemic apoE-deficient mice which develop massive atherosclerosis to test plaque stability after administering RvT4. ORO staining is also used for evaluating aortic lipid load but atherosclerosis extension is not measured. It is unclear why the authors evaluated the collagen content in brachiocephalic arteries, but not in the proximal aorta (more related with aortic lipid load).

8. In different experiments, the number of mice is very low (3 per group). I do have concerns about using a student t-test with 3 individual subjects to get conclusions.

Reviewer #2 (Remarks to the Author):

Walker and colleagues explored the potential of Resolvin T4 to limit vascular and joint inflammation using mouse models and macrophages. They conclude that resolvin T4 helps

to remove cholesterol from lipid-laden macrophages, and therefore reduces vascular lesions in aorta and lowers joint inflammation.

The topic is novel, original and the manuscript describes potentially very interesting properties of a signaling lipid. However, I have several comments to the data analysis, presentation and statistics.

Major points:

1. The manuscript connects two stories – one focused on joint inflammation and one on atherosclerosis – and the story line is not balanced and clear. Results from both parts are mixed together to fit the narrative, but I would suggest to remove the arthritis part and keep only data on atherosclerosis, macrophages and the mechanisms for clarity.
2. Figure 1 – I do not understand the rationale behind selection of RvT4 for the experiments. According to Fig.1E, RvT4 is significant, but RvT1 shows much stronger decrease. Also, the RvT1 p value is 0.061, but if I calculate (approximation) the shown values with a t-test, I get a significant result. Please explain and correct. Why RvT1 was not selected?
3. Figure 1 – Please clarify the number of mice per experiment. Fig.1E shows 4 point, but n=3 is mentioned in the legend. Fig.1F has 10 and 13 point. Fig.1G has 12 points. Fig.G legend is missing.
4. Figure 1 – statistics (and other figures as well). I think that n=3 for mouse experiment is underestimated. There is no power calculation presented within Statistics section. Many bar graphs are missing error bars, n is undefined or low. Test like subjective scoring of mouse joint pain using n=3 is not convincing. And it was performed only once (?).
5. Figure 2D – PSD-DA is supervised/biased and you present only 3 mice per group. Also, the list of input metabolites is missing. If the list of metabolites is identical to Table S1 content, is it also biased, because there are only related metabolites at very low levels. Please provide PCA evaluation.
6. Figure 2E. There is no variability (error bar) shown for vehicle. The same for Fig.3E. Please provide data.
7. Figure 4AB- Why a combination of 2w ANOVA and t-test was used here? Fig.4B 1nM value is not significantly changed? It is not very convincing – was ANOVA used for the dependent variables?
8. Figure 4C – No error bar is shown for vehicle. Does it mean that either no cells migrated in

control group or the same number (unknown?) migrated in all control samples? Then, what is means that ~50% or ~180% of control group migrated. Please explain in the text and clarify.

9. Figure 5B – Please expand the Y axis. This is more just a trend.

10. Figure 5C – Statistics is unclear. If ANOVA was used, there is no difference between 0 a 10 minutes? Also the cholesterol quantity should be clarified. Usually, free cholesterol is orders of magnitude more concentrated than CEs. Is this true for oxLDL-loaded cells? It should be shown how much of (free and bound) cholesterol in in the cells and in media to be able interpret the real efflux effect.

11. Figure 6B – again, no error bar is shown for Vehicle and n=3. Such result can't be properly reviewed. Please explain "n=3 per group from 2 distinct experiments". Does it means you used 3 volunteers, prepared 12 wells for the experiment and repeated this twice?

12. Figure 6I – This style is inconsistent with the other plots and unclear.

Reviewer #3 (Remarks to the Author):

The data related to the role of a new resolvin in macrophage biology and inflammation is novel and interesting. A major concern relates to the reliance on very few arthritis experiments with limited numbers of animals. As a general rule, most murine arthritis models require at least 5-6 mice per group and experiments are performed at least twice. Reliance on data with only 3 animals per group with no repeats is highly problematic and are features of Figure 1, 3, 6, and 7.

Other concerns for Figure 1:

1. The difference between control and WD is very small. Differences less than ~30% are often considered statistically significant but not clinically significant.
2. Figure 1b exaggerates the effect by using a y axis that begins at 120 instead of 0. This is an issue with many of the figures.
3. Conclusions related to cartilage damage are not supported because of the low number and the overlap between the groups. The authors imply that it is significant.
4. There is also limited information on common endpoints in these models (eg cytokines in

the joint? Standard histopathology (that was done in figure 3)?)

5. Part F is interesting but not convincing

6.

Furthermore, there are no data showing that RvT4 plays a role in the MOA of WD, and the authors rely on correlations. This figure is used as a rationale to subsequent studies, but it would need more experiments and an intervention with WD to block RvT4.

Figure 3 shows a somewhat more convincing effect, but the use of Ri is misleading as an outcome. The measure that they use is the time from peak arthritis to a fixed amount of arthritis score. If the rate of arthritis resolution was the same, the group starting with lower peak score would look like they have faster resolution. A much better way to analyze rate of resolution this is to use deltas or percent change. This is also a problem with Figure 7a. Again, the number of mice and replicates is insufficient to draw conclusions. The conclusions about the role of RvT4 are mainly due to associations rather than directly showing that WD acts through this mechanism (eg, WD exacerbate disease, WD has lower RvT4, RvT4 treatment decreases arthritis, therefore RvT4 is responsible for WD effects).

Figure 6 uses an interesting ex vivo model, but it is not clear why these studies were not performed directly on cells isolated from joints during arthritis in mice treated with or without RvT4. The hypothesis related to changes in MerTK macrophages can be experimentally in the arthritic mice by flow cytometry, immunostaining, or other techniques on joint tissues directly from the model rather than requiring ex vivo culture.

Overall, the concepts are very interesting but the study has major flaws:

1. Inadequate number of mice and replicates, with overstating the results
2. Lack of direct evidence showing that RvT4 is the mechanism of WD effects

Reviewer #1

This study investigated the effect of RvT4 on macrophage lipid accumulation, joint inflammation and aortic atherosclerosis. The authors report that RvT4 may promote cholesterol efflux in foam cells via SR-BI, thereby reprogramming lipid laden macrophages and yielding tissue protection. Although I find the study novel and interesting and of potential wide interest, some major conclusions are not supported by the data, and the study of lipoproteins (including HDL, cholesterol efflux and reverse cholesterol transport) and atherosclerosis are in need of a major improvement.

We thank this reviewer for their focus and expertise on cholesterol efflux, macrophages, and atherosclerosis. We are also thankful to the reviewer for highlighting aspects related to dissecting the mechanism of action of RvT4 that we had not previously addressed. These suggestions, along with the advice related to aorta lesion quantification and contents of the aortic root, have further supported the role for RvT4 in macrophage cholesterol efflux through SR-BI, culminating in improved atherosclerotic outcomes.

1. The effects of RvT4 in vivo on plasma lipids and lipoproteins should be shown (free and total cholesterol, triglycerides, and cholesterol in plasma lipoprotein fractions) in mice studies.

We thank the reviewer for these insightful suggestions. We have performed additional experiments to address this comment demonstrating that RvT4 increased the ratio of HDL:Total cholesterol, reduced plasma triglyceride concentrations and reduced plasma phospholipid concentrations in ApoE^{-/-}. These findings are presented in figure 7 I-J and discussed on page 11 lines 311-313 of the revised manuscript as detailed below:

“This vascular protective activity of RvT4 in ApoE^{-/-} mice was linked with a significant upregulation of plasma HDL to total cholesterol ratio and a downregulation of plasma triglyceride and phospholipid concentrations (Figure 7H-J).”

2. The idea that RvT4 is affecting macrophages SR-BI only in articulations and vasculature needs to be proven. More specifically, authors should rule out that RvT4 is also affecting liver SR-BI, where it would have the potential to induce major changes in circulating HDL levels.

Thank you for this suggestion. To address the role of RvT4 on SR-BI expression in the liver, ApoE^{-/-} mice were injected with RvT4 intravenously on days 0 and 2. On day 3, the liver was perfused to remove circulating leukocytes and a single-cell suspension prepared. The expression of SR-BI on liver cells was measured using flow cytometry. Results from these experiments demonstrate that RvT4 does not significantly regulate SR-BI expression on hepatocytes. These findings are presented in Figure 6G and discussed on page 10 lines 274-280 of the revised manuscript as detailed below:

“SR-BI plays a role in cholesterol uptake from HDL in the liver (Rigotti et al., 1997). Thus, we next evaluated whether the protective activities observed for RvT4 in vivo were also mediated via the regulation of hepatic SR-BI expression. For this purpose, mice were injected with RvT4 and SR-BI expression in liver cells was measured using flow cytometry. Results from these experiments demonstrate that expression of this receptor was not significantly regulated by RvT4 on hepatic cells (Figure 6G). Thus, these findings support the hypothesis RvT4 selectively regulated SR-BI expression in macrophages.”

3a. Figure 5 data would seem critical for concluding that RvT4 is promoting macrophage cholesterol efflux via SR-BI. However, the authors did not determine cholesterol efflux. Only in panel A, the authors demonstrated that RvT4 promoted cholesterol release into the medium, but the rest of panels are determining macrophage neutral lipid accumulation, not the ability of these macrophages to promote macrophage cholesterol efflux. The legend is difficult to follow. For example, in panel I the authors indicated that macrophages were incubated with ox-LDL then with 1nM RvT4 and lipid efflux was assessed; but lipid efflux is not shown (only neutral lipid load).

Thank the reviewer for raising these important points. We have performed additional experiments assessing extracellular cholesterol levels *in vitro* and *in vivo* to support the conclusion that the observed reduction in macrophage lipid load was due to reverse cholesterol transport. These new experiments are presented in Figures 4E, 5E, 5G and 5J and discussed on page 8 lines 211-220 and page 9 lines 242-250 as detailed below. We have also revised the figure legend of Figure 4 splitting this into two figures (Figures 4 and 5) to enhance clarity.

Page 8 lines 211-220

“We next sought to evaluate whether our in vitro findings were translatable to the in vivo scenario. For this purpose, a fluorescently labelled cholesterol was delivered into the peritoneal cavity of ApoE^{-/-} mice. These animals were then treated with RvT4 and intracellular cholesterol load in peritoneal macrophages was evaluated using flow cytometry. This experiment demonstrated a significant reduction in intracellular cholesterol levels in cells recovered from the peritoneum of mice treated with RvT4 when compared with mice that received vehicle alone (Figure 4D). This decrease in intracellular cholesterol was coupled with a significant increase in extracellular cholesterol measured in cell free peritoneal lavages from RvT4 treated mice (Figure 4E). Together these findings lend support to the hypothesis that RvT4 regulates reverse cholesterol transport in macrophages.”

Page 9 lines 242-250

“Indeed, knockdown of ABCA1 or ABCG1 expression did not limit the ability of RvT4 to promote the release of cholesterol from macrophages (Figure 5D, E). Incubation of these cells with BLT-1, an inhibitor to SR-BI, abolished the ability of RvT4 to promote macrophage cholesterol efflux as denoted by an ablation of RvT4 to limit both intracellular lipid load and total cholesterol levels in the supernatants (Figure 5F, G). Additionally, knockdown of SR-BI using an siRNA against its gene SCARB1, also blunted the ability of RvT4 to reduce macrophage neutral lipid load and increase cholesterol efflux (Figure 5H-J).”

3b Which are the acceptors used in the different experiments: whole plasma, isolated HDL?

Experiments performed in Figure 4A-C and Figure 5 and Figure 6 were performed in PBS without the addition of serum, plasma or isolated HDL. Whereas in experiments presented in Figure 6D-E isolated HDL was used. As can be observed from the results presented in these figures, the effects of RvT4 in regulating RCT in macrophages *via* SR-B1 occurred in the presence and absence of an acceptor.

3c There is a need of studying cholesterol efflux directly by using standard methodology. There are ways to distinguish the efflux ABCA1 or ABCG1 dependent that will need to be performed.

To further evaluate the role of each of these transporters in mediated the observed activities of RvT4 we performed siRNA mediated KD in human monocyte-derived macrophages and assessed the impact of their downregulation on the ability of RvT4 to downregulate macrophage lipid load. Results from these studies are reported in Figure 5D and E and discussed on page 9 lines 242-250 as detailed below:

“Indeed, knockdown of ABCA1 or ABCG1 expression did not limit the ability of RvT4 to promote the release of cholesterol from macrophages (Figure 5D, E).”

4. I do not think that macrophage SR-B1 is currently considered a major mechanism for cholesterol efflux. In fact it is usually included on the term "unspecific efflux". Since ABCA1 and ABCG1 are in contrast critical in specific cholesterol efflux, they should be directly determined both *in vitro* and *in vivo*.

We refer the reviewer to the response to point 3c above demonstrating that knockdown of these two transporters *in vitro* does not limit the ability of RvT4 to increase extracellular cholesterol levels. To further support the role of macrophage SR-B1 in mediating the vasculo-protective activities of RvT4 *in vivo* we performed additional experiments ablating the expression of this receptor in macrophages. Results from these studies demonstrate that ablation of this receptor on macrophages completely reverses the protective activities of this SPM *in vivo* in *ApoE*^{-/-} mice. Results from these experiments are presented in Figure 7E-J and discussed on page 11 lines 300-317 as detailed below:

*“Having observed that *in vitro* SR-B1 expression was rapidly regulated by RvT4, a mechanism that was linked with the ability of SPM to regulate RCT in macrophages, we next questioned whether this mechanism was responsible for the vasculo-protective activities of RvT4. For this purpose, we used a macrophage targeting siRNA approach to abrogate the expression of this receptor on macrophages in *ApoE*^{-/-} mice fed a WD (see Figure 7D). The effectiveness of the approach was corroborated by flow cytometry which highlighted a significant downregulation of SR-B1 expression in aortic macrophages from mice treated with siRNA targeting *Scarb-1* when compared with mice treated with a control sequence (Figure 7E). Quantification of vascular lipid load demonstrated that RvT4 given to mice treated with a control siRNA sequence significantly reduced vascular lesions and decreased necrotic core area (Figure 7F,G). Intriguingly the ability of RvT4 to limit vascular inflammation was abrogated in mice treated with siRNA targeting macrophage *Scarb-1* (Figure 7F). This vascular protective activity of RvT4 in *ApoE*^{-/-} mice was linked with a significant upregulation of plasma HDL to total cholesterol ratio and a downregulation of plasma triglyceride and phospholipid concentrations (Figure 7H-J). Notably the ability of RvT4 to regulate plasma lipid levels was abrogated in mice that received an siRNA targeting *Scarb-1* expression in macrophages (Figure 7H-J). Taken together, these findings support a role for macrophage SR-B1 in mediating the vasculo-protective properties of RvT4 during inflammatory arthritis (Figure 7F).”*

5. Cholesterol efflux is only the first RCT step. Therefore, the authors need to show in mice that RvT4 promotes increase in macrophage-specific reverse cholesterol transport. This would be the physiological mechanism that could explain, at least in part, atherosclerosis improvement.

To address this comment we loaded peritoneal macrophages of ApoE^{-/-} mice with cholesterol by injecting BODIPY-FL-labelled cholesterol into the peritoneal cavity and the ability of RvT4 to reduce macrophage lipid load and increase the release of cholesterol into the peritoneum was evaluated. Results from these experiments demonstrate, that as observed *in vitro*, RvT4 decreases macrophage lipid load and increases extracellular cholesterol levels in line with the hypothesis that this SPM regulates RCT. Results from these experiments are presented in Figure 4D,E and discussed on page 8 lines 211-220 of the revised manuscript as detailed below:

“We next sought to evaluate whether our in vitro findings were translatable to the in vivo scenario. For this purpose, a fluorescently labelled cholesterol was delivered into the peritoneal cavity of ApoE^{-/-} mice. These animals were then treated with RvT4 and intracellular cholesterol load in peritoneal macrophages was evaluated using flow cytometry. This experiment demonstrated a significant reduction in intracellular cholesterol levels in cells recovered from the peritoneum of mice treated with RvT4 when compared with mice that received vehicle alone (Figure 4D). This decrease in intracellular cholesterol was coupled with a significant increase in extracellular cholesterol measured in cell free peritoneal lavages from RvT4 treated mice (Figure 4E). Together these findings lend support to the hypothesis that RvT4 regulates reverse cholesterol transport in macrophages.”

6. In the manuscript, atherosclerosis area is quite more extensive than usual under Western diet. Wild-type mice usually develop early fatty streak under a Western-type diet and this is mainly located in proximal aorta. ORO staining intensity is not a relevant measure of atherosclerosis extension (rather the % of area covered by lesions).

We thank the reviewer for this suggestion. We revised the analytical approach assessing percentages of area covered by lesions. Results from these analyses are presented in Figure 2B and Figure 7F of the revised manuscript and discussed on page 6 lines 134-136 and page 11 lines 307-309 as detailed below:

page 6 lines 134-136

“Using oil-red O stain we observed a significant reduction in vascular lipid load in the aortic arch of arthritic mice fed a WD and treated with RvT4 (Figure 2B).”

Page 11 lines 307-309

“Quantification of vascular lipid load demonstrated that RvT4 given to mice treated with a control siRNA sequence significantly reduced vascular lesions and decreased necrotic core area (Figure 7F,G).”

7a. In figure 7, the authors used hyperlipidemic apoE-deficient mice which develop massive atherosclerosis to test plaque stability after administering RvT4. ORO staining is also used for evaluating aortic lipid load but atherosclerosis extension is not measured.

We have performed additional experiments evaluating ORO staining in the descending aorta in addition to the aortic arch. Results from these experiments are presented in figure 7F as detailed above.

7b. It is unclear why the authors evaluated the collagen content in brachiocephalic arteries, but not in the proximal aorta (more related with aortic lipid load).

The selection of this tissue is based on published literature demonstrating that atherosclerosis extends to this vessel. Based on the reviewer's remark we removed the original results from the manuscript and performed additional experiments assessing necrotic core area in the aortic roots from mice treated with or without RvT4. Unfortunately, due to technical reasons we were unable to evaluate collagen content in these tissues too. The results obtained evaluating necrotic core area in the aortic roots are presented in Figure 7G and discussed on page 11 lines 411-413:

“Quantification of vascular lipid load demonstrated that RvT4 given to mice treated with a control siRNA sequence significantly reduced vascular lesions and decreased necrotic core area (Figure 7F,G).”

8. In different experiments, the number of mice is very low (3 per group). I do have concerns about using a student t-test with 3 individual subjects to get conclusions.

We thank the reviewer and understand the nature of these comments. We had previously carried out power calculations to determine the number of subjects required to observe statistical significance (achieving a power of 80%) in each given experiment, and the value of 3 was sufficient. Nonetheless in order to address this genuine concern, we have performed additional experiments to increase *n* numbers where we tested the effects of RvT4 in disease systems. In Figure 2B, where arthritic wild-type mice fed Western diet were given vehicle or RvT4, and in Figure 6B, where arthritic *ApoE*^{-/-} mice fed a Western diet were given vehicle or RvT4, the *n* was increased to 7 mice per group. In addition, in experiments performed as part of the revisions including results presented in Figure 4, 5, 6 and 7 we use *n* number greater than 4 for *in vitro* experiments and greater than 7 for *in vivo* experiments.

Reviewer #2

Walker and colleagues explored the potential of Resolvin T4 to limit vascular and joint inflammation using mouse models and macrophages. They conclude that resolvin T4 helps to remove cholesterol from lipid-laden macrophages, and therefore reduces vascular lesions in aorta and lowers joint inflammation.

The topic is novel, original and the manuscript describes potentially very interesting properties of a signaling lipid. However, I have several comments to the data analysis, presentation and statistics.

We thank the reviewer for their complimentary comments.

Major points:

1. The manuscript connects two stories – one focused on joint inflammation and one on atherosclerosis – and the story line is not balanced and clear. Results from both parts are mixed together to fit the narrative, but I would suggest to remove the arthritis part and keep only data on atherosclerosis, macrophages and the mechanisms for clarity.

We thank the reviewer for this suggestion, and in accord with the editor's recommendations, we have modified the manuscript which now focuses on atherosclerosis and macrophages.

2. Figure 1 – I do not understand the rationale behind selection of RvT4 for the experiments. According to Fig.1E, RvT4 is significant, but RvT1 shows much stronger decrease. Also, the RvT1 p value is 0.061, but if I calculate (approximation) the shown values with a t-test, I get a significant result. Please explain and correct. Why RvT1 was not selected?

We thank the reviewer for this comment. Figure 1E does indeed present a strong decrease in the concentration of RvT1 compared to RvT4. Of these two mediators, only RvT4 demonstrated a significant correlation with vascular lipid load as presented in Figure 1F. It is for this reason that we opted to focus on this mediator. To clarify our rationale, we have included the correlation results for RvT1 concentrations with vascular lipid load in the revised manuscript which were previously omitted due to space restrictions (Figure 1G). With respect to statistical calculations, we have reviewed the analysis and can confirm that using the described test the p value is of 0.06.

3. Figure 1 – Please clarify the number of mice per experiment. Fig.1E shows 4 point, but n=3 is mentioned in the legend. Fig.1F has 10 and 13 point. Fig.1G has 12 points. Fig.G legend is missing.

We thank the reviewer for highlighting these typos. We have revised these and ensured that the correct values are displayed in the legends throughout the manuscript.

4. Figure 1 – statistics (and other figures as well). I think that n=3 for mouse experiment is underestimated. There is no power calculation presented within Statistics section. Many bar graphs are missing error bars, n is undefined or low. Test like subjective scoring of mouse joint pain using n=3 is not convincing. And it was performed only once (?).

The n numbers were determined using power calculations, unfortunately this important detail was erroneously omitted from the submitted version of the manuscript. Nonetheless we agree with the reviewer on the issue of replication and have endeavoured to increase the n numbers wherever possible for both *in vitro* and *in vivo* experiments. N numbers have been increased for experiments presented in Figures 2B, 2D, 2E, 4D-E, and 7. We have also included a reference to the power calculations in the statistics section within the methods on page 32 lines 810-813 of the revised manuscript as detailed below:

“The number of subjects used in each experiment was determined by carrying out power calculations using historical data presenting strong effect sizes, to reach a power of 80%. In these experiments, the given number of subjects was used, or a minimum of n=3 per group was used to reduce animal use.”

5. Figure 2D – PSD-DA is supervised/biased and you present only 3 mice per group. Also, the list of input metabolites is missing. If the list of metabolites is identical to Table S1 content, is it also biased, because there are only related metabolites at very low levels. Please provide PCA evaluation.

We thank the reviewer for this comment. We have increased the n number and performed PCA analysis as requested. Results from the revised analysis demonstrate that the shift in LM profiles observed previously is retained even when using an unsupervised analysis. These results are presented in Figure 2D of the revised manuscript and discussed on page 6 lines 141-146 as detailed below:

“Whereby principal component analysis (PCA) unveiled a shift in the SPM concentrations in aortas from mice treated with RvT4 when compared with mice treated with vehicle alone. This was demonstrated by a leftward shift in the cluster representing these mice, when compared with mice receiving vehicle. This separation in mediator profiles was linked with the upregulation of several pro-resolving mediators including 15-epi-lipoxin A4 and RvD5 (Figure 2D and Supplemental Table 1).”

6. Figure 2E. There is no variability (error bar) shown for vehicle. The same for Fig.3E. Please provide data.

The reason why there was no variability in the data was because we evaluated the differences between the mean value for the vehicle group to that observed in mice treated with RvT4. We have revised this analysis to report the individual data points. The revised figure is presented in Figure 2E and discussed on page 6 lines 153-155 of the revised manuscript as detailed below. As suggested by the reviewer and instructed by the editor, the data relating to the role of RvT4 in regulating joint inflammation (i.e. Figure 3E) have been removed from the revised manuscript.

‘Assessment of lipid load in macrophages demonstrated a significant reduction in lipid content, as measured using LipidTOX staining, when RvT4 was added to these organ cultures (Figure 2E).’

7. Figure 4AB- Why a combination of 2w ANOVA and t-test was used here? Fig.4B 1nM value is not significantly changed? It is not very convincing – was ANOVA used for the dependent variables?

We thank the reviewer for bringing this typo to our attention. The bar highlighting significance between vehicle and 1nM RvT4 was misplaced and has been rectified (Figure 4B). The authors confirm that an ANOVA was used for these statistical calculations. The figure (Figure 3B) and corresponding figure legend have been revised.

8. Figure 4C – No error bar is shown for vehicle. Does it mean that either no cells migrated in control group or the same number (unknown?) migrated in all control samples? Then, what it means that ~50% or ~180% of control group migrated. Please explain in the text and clarify.

As detailed above, the values presented in our original submissions presented the average value of the respective control group, representing 100%. Therefore the 50% and 180% values reported were the % change from this average value. To enhance the clarity of the manuscript we have revised the figure to report the raw data. This data is now reported in Figure 3 of the revised manuscript.

9. Figure 5B – Please expand the Y axis. This is more just a trend.

We have expanded the y-axis for this figure as requested. The revised figure is now presented in Figure 4B.

10. Figure 5C – Statistics is unclear. If ANOVA was used, there is no difference between 0 and 10 minutes? Also the cholesterol quantity should be clarified. Usually, free cholesterol is

orders of magnitude more concentrated than CEs. Is this true for oxLDL-loaded cells? It should be shown how much of (free and bound) cholesterol in the cells and in media to be able interpret the real efflux effect.

We thank the reviewer for bring this error to our attention, the p value was omitted from this Figure during manuscript preparation. This mistake is rectified in the revised Figure 4C.

With regards to the concentrations of free cholesterol and cholesteryl esters within the cells, we note that cholesterol in the cell is typically found as cholesteryl esters due to the toxic nature of excessive free cholesterol. Figures 4B and 4C, in combination, indicate that as incubation time increases, the amount of total cholesterol decreases but the percentage of free cholesterol increases. These findings suggest that there is an increasing amount of free cholesterol within the cell, which is the form which can be transported by SR-BI to the extracellular space. This mechanism is supported by results presented in Figure 4A, where incubation of macrophages with RvT4 increases free cholesterol in cell free supernatants. We also provide additional results relating to extracellular cholesterol levels in Figures 4E, 5E, 5G and 5J.

11a. Figure 6B – again, no error bar is shown for Vehicle and n=3. Such result can't be properly reviewed.

We refer the reviewer to the response to point 6 above in regards to the lack of error bar in the vehicle group. To include error bars as suggested, we have removed data normalisation such that the raw fluorescence intensities are presented (Figures 4D-H, 4K, 5B-D). ANOVA was then used to carry out statistics in these experiments, with the appropriate multiple comparisons test.

11b. Please explain “n=3 per group from 2 distinct experiments”. Does it means you used 3 volunteers, prepared 12 wells for the experiment and repeated this twice?

In this instance (Figure 5B), a total of 3 donors were used over 2 distinct experiments. In the first experiment, cells from one donor were used to test whether RvT4 inhibited SR-B1 translocation, i.e. evaluating cellular responses to all four conditions from the same donor. In the second experiment which was performed on a separate day, we used cells from two additional donors to evaluate the same cellular responses.

12. Figure 6I – This style is inconsistent with the other plots and unclear.

This panel has been removed as part of the arthritis section, to streamline the paper. We however thank the reviewer for this suggestion, which will be implemented in the future.

Reviewer #3

The data related to the role of a new resolvin in macrophage biology and inflammation is novel and interesting. A major concern relates to the reliance on very few arthritis experiments with limited numbers of animals. As a general rule, most murine arthritis models require at least 5-6 mice per group and experiments are performed at least twice. Reliance on data with only 3 animals per group with no repeats is highly problematic and are features of Figure 1, 3, 6, and 7.

We thank the reviewer for these comments and understand the limitations of n=3. The results pertaining to the role of Western diet of the severity of arthritis (Figure 1) has been presented by other groups such as the following: <https://link.springer.com/article/10.1186/ar4261> and these results were only used to stage the findings presented in the later panels. Given the minimal variation in the effect sizes, the presence of historical data from different groups, and power calculations carried out prior to the experiments suggesting that n=3 was sufficient to give a power of 80%, n=3 was maintained. However, to streamline the presentation of our study, and in accord with the editor's request we have omitted all data related to the role of RvT4 in regulating joint inflammation.

Other concerns for Figure 1:

1. The difference between control and WD is very small. Differences less than ~30% are often considered statistically significant but not clinically significant.

We thank the reviewer for this comment. It should be noted that assessment of the area under the curve for clinical scores gives an increase in clinical disease severity of ~28%. Thus, as also shown by other groups, the role of Western diet on prolonged K/BxN arthritis can be deemed clinically significant.

2. Figure 1b exaggerates the effect by using a y axis that begins at 120 instead of 0. This is an issue with many of the figures.

In line with the reviewer's suggestion, we have revised Figure 1b and other figures starting the y-axis at 0.

3. Conclusions related to cartilage damage are not supported because of the low number and the overlap between the groups. The authors imply that it is significant.

This data has been omitted from the revised manuscript.

4. There is also limited information on common endpoints in these models (eg cytokines in the joint? Standard histopathology (that was done in figure 3)?)

We concord with the reviewer on the utility of having additional endpoints commonly used to evaluate joint inflammation as additional outcomes. However, since we were requested to exclude all outcomes related to joint inflammation, these were not included in the revised manuscript.

5. Part F is interesting but not convincing

As discussed above, all joint related data, included Figure 3F in the original submission was omitted from the revised manuscript.

6. Furthermore, there are no data showing that RvT4 plays a role in the MOA of WD, and the authors rely on correlations. This figure is used as a rationale to subsequent studies, but it would need more experiments and an intervention with WD to block RvT4.

We believe that there is a misunderstanding for the above comment. Indeed, Figure 1 demonstrates that in arthritic mice fed a Western diet (i.e. an intervention with WD) there are lower concentrations of RvT4 than in arthritic mice fed a standard chow diet (Figure 1E). This suggests to us that WD limits RvT4 production. This is further supported by data presented in Supplemental figure 1 which demonstrates a downregulation in the products of

the RvT biosynthetic enzymes ALOX5 and COX2, 7-HDPA and 13-HDPA, respectively, in mice fed a WD.

Figure 3 shows a somewhat more convincing effect, but the use of Ri is misleading as an outcome. The measure that they use is the time from peak arthritis to a fixed amount of arthritis score. If the rate of arthritis resolution was the same, the group starting with lower peak score would look like they have faster resolution. A much better way to analyze rate of resolution this is to use deltas or percent change. This is also a problem with Figure 7a. Again, the number of mice and replicates is insufficient to draw conclusions. The conclusions about the role of RvT4 are mainly due to associations rather than directly showing that WD acts through this mechanism (eg, WD exacerbate disease, WD has lower RvT4, RvT4 treatment decreases arthritis, therefore RvT4 is responsible for WD effects).

We thank the reviewer for this suggestion. As detailed above this data has been omitted from the revised manuscript upon editorial request.

Figure 6 uses an interesting ex vivo model, but it is not clear why these studies were not performed directly on cells isolated from joints during arthritis in mice treated with or without RvT4. The hypothesis related to changes in MerTK macrophages can be experimentally in the arthritic mice by flow cytometry, immunostaining, or other techniques on joint tissues directly from the model rather than requiring ex vivo culture.

We thank the reviewer for these intriguing suggestions which we will explore in future experiments.

Overall, the concepts are very interesting but the study has major flaws:

1. Inadequate number of mice and replicates, with overstating the results
2. Lack of direct evidence showing that RvT4 is the mechanism of WD effects

We have addressed the above points by increasing the number of mice per group for the as outlined in the responses to reviewers 1 and 2 and also by assessing the role of macrophage SR-BI in mediating the protective activities of RvT4 in vivo. We also agree with the reviewer that it would be of interest to establish how WD influences the activity of RvT4 biosynthetic enzymes however we feel that this aspect is beyond the scope of the present manuscript and will be addressed in future studies. The authors would like to thank this reviewer for the helpful comments relating to arthritis. Although we were required to omit the data related to the joint inflammation outcomes these will be included in our future studies detailing these outcomes.

REVIEWER COMMENTS

Reviewer #1 (Remarks to the Author):

The manuscript has improved substantially and most of the reviewers' concerns have been addressed. However, some points need to be considered further:

- 1) In vivo macrophage reverse cholesterol transport evaluation need be completed by measuring labeled cholesterol in feces (usually after 72 h of labeled cholesterol-loaded macrophage injection into the peritoneal cavity of mice). This is the end of the in vivo pathway and the result is of interest.

- 2) RvT4 in vivo on plasma induces a significant increase in the ratio of HDL cholesterol:Total cholesterol whereas reducing plasma triglyceride levels. Is the reduction in plasma HDL:total cholesterol ratio in apoE-deficient mice due to changes in total serum cholesterol levels, HDL cholesterol, or both? Could you include absolute plasma concentrations of each?

- 3) Please review the figure 6 C and D, and 5A and 5B, as the distribution of points looks similar thus questioning the significance reported.

- 4) The contribution of macrophage SR-B1 to the pathogenesis of atherosclerosis is dependent of the stage of foam cell formation (reviewed in *Atherosclerosis* 2017;258:153-161) and you may consider this topic as a element for discussion. Further, there is a recent review on SR-B1 wich you might consider as an additional reference (Yu L, Day Y, Mineo C. *Curr Atheroscler Rep* 2021).

Reviewer #2 (Remarks to the Author):

The present revision shows good progress in clarification of the story. My comments were addressed and new experiments in vivo and in vitro were performed.

However, the statistics continues to be the major problem. Several types of ANOVA are used incorrectly. As the p-values are borderline and the number of observations is low, robust statistics has to be performed to support the data.

Line 145-146, PCA - please provide statistical details in the SI (PCA scores, loadings, model parameters)

Line 175, Figure 3A - how is the output concentration-dependent? The two other concentrations 0.1 and 10 nM are neither mentioned nor discussed. How is this experiment output evaluated using 2-way ANOVA?

Figure 5 - The statistics is unclear. Panel G is a typical situation for 2-way ANOVA and the interactions, but 1-way ANOVA is mentioned at the end of the legend. This should be consulted with a statistician, appropriate tests should be used, and only the hypotheses tested by these tests should be discussed.

Figure 6 - The same as above. Panel C should be evaluated using 2-way ANOVA, etc.

Minor points:

There are many typos in the new (yellow) text.

Figure 4B, please reformat the units

Reviewer #1 (Remarks to the Author):

The manuscript has improved substantially and most of the reviewers' concerns have been addressed. However, some points need to be considered further:

1) In vivo macrophage reverse cholesterol transport evaluation need be completed by measuring labeled cholesterol in feces (usually after 72 h of labeled cholesterol-loaded macrophage injection into the peritoneal cavity of mice). This is the end of the in vivo pathway and the result is of interest.

The authors would like to thank the reviewer for this suggestion. Using fluorescently labelled cholesterol loaded murine macrophages we evaluate the ability of RvT4 to regulate fecal cholesterol levels. Results from these studies demonstrated an upregulation of fecal cholesterol levels at the 72h time point. Results from these experiments are presented in figure 6G and discussed on page 10 lines 277-283 of the revised manuscript as detailed below:

The last step in RCT is the excretion of HDL-associated cholesterol via the liver. Thus, we next evaluated whether RvT4 increased fecal levels of macrophage-derived cholesterol. To evaluate this we transferred macrophages loaded with fluorescently labelled cholesterol into the peritoneum of ApoE^{-/-} mice and quantified the fecal fluorescence levels after 72 hours. Here we observed a marked increase in fecal associated cholesterol in mice treated with RvT4 when compared with those that received vehicle alone, supporting the hypothesis that RvT4 promoted RCT (Figure 6G).

2) RvT4 in vivo on plasma induces a significant increase in the ratio of HDL cholesterol:Total cholesterol whereas reducing plasma triglyceride levels. Is the reduction in plasma HDL:total cholesterol ratio in apoE-deficient mice due to changes in total serum cholesterol levels, HDL cholesterol, or both? Could you include absolute plasma concentrations of each?

We have included the requested data in Supplemental Table 3. These findings demonstrate that the observed increase was primarily driven by a reduction of Total serum cholesterol levels which was linked with a marked reduction in plasma LDL associated cholesterol.

3) Please review the figure 6 C and D, and 5A and 5B, as the distribution of points looks similar thus questioning the significance reported.

Thank you for bringing this to our attention. We have reviewed the raw data underlying these figures and can confirm their significance. The spread in the datasets is linked with an intra-donor variability.

4) The contribution of macrophage SR-B1 to the pathogenesis of atherosclerosis is dependent of the stage of foam cell formation (reviewed in *Atherosclerosis* 2017;258:153-161) and you may consider this topic as a element for discussion. Further, there is a recent review on SR-B1 wich you might consider as an additional reference (Yu L, Day Y, Mineo C. *Curr Atheroscler Rep* 2021).

We would like to thank the reviewer for suggesting these references to use. We found them insightful and thought-provoking. We included these as references on page 15, line 409-412 and lines 421-422 as detailed below:

page 15, line 409-412

“For many years the role of SR-BI in regulating cholesterol homeostasis and protecting against vascular inflammation was primarily linked with uptake of cholesterol esters in the liver. Recent findings have challenged this hypothesis, indicating the role of SR-BI in mediating cholesterol efflux from macrophages to mediate atherosclerosis”.

page 15, line 421-422

“Endothelial SR-BI mediates the transport of both HDL and LDL to the subintimal space to modulate the extent of atherosclerosis”

Reviewer #2 (Remarks to the Author):

The present revision shows good progress in clarification of the story. My comments were addressed and new experiments in vivo and in vitro were performed. However, the statistics continues to be the major problem. Several types of ANOVA are used incorrectly. As the p-values are borderline and the number of observations is low, robust statistics has to be performed to support the data.

Line 145-146, PCA - please provide statistical details in the SI (PCA scores, loadings, model parameters)

We included a Supplementary Figure 4 (page 4, Supplemental Figures and Tables file) containing the loading plot, model parameters, and PCA scores.

Line 175, Figure 3A - how is the output concentration-dependent? The two other concentrations 0.1 and 10 nM are neither mentioned nor discussed. How is this experiment output evaluated using 2-way ANOVA?

Specialized pro-resolving mediators typically display a bell-shaped dose-response curve. This in this experiment we evaluate three separate concentrations of RvT4 observing that the 1nM concentration displayed the greatest potency at governed key biological activities in the regulation of vascular inflammation (page 7, line 174-189) as detailed below:

“Assessment of apoptotic cell uptake in real-time using high content imaging demonstrated that the presence of RvT4 increased both the rate at which lipid-laden macrophages engulfed apoptotic cells as well as their overall capacity to ingest these cells (Figure 3A,B). Thirty minutes after the addition of apoptotic cells, lipid-laden macrophages in the 1nM group had taken up an additional 131% apoptotic cells compared to vehicle, whereas lipid-laden macrophages in the 0.1nM and 10nM groups engulfed an additional 54% and 90% respectively (Figure 3A).

Lipid laden macrophages also display an impaired ability to migrate, resulting in these cells becoming trapped in tissues and perpetuating local inflammation (Doran et al., 2020; Libby et al., 2014). Thus, we next assessed whether RvT4 also improved macrophage chemotaxis. While incubation of lipid laden macrophages with RvT4 alone did not increase migration in the absence of a chemoattractant (Figure 3C), cell migration towards two classic chemoattractants, MCP-1 (Figure 3D) and ATP (Figure 3E), was increased by addition of RvT4 in a concentration-dependent manner. These results indicate that RvT4 restores the ability of lipid-laden macrophages to respond to a chemotactic queue.’

We additionally included a line on the rationale for using 1nM RvT4 in subsequent experiments (page 7, lines 191-194) as detailed below:

“To establish if this increased chemotaxis was observed in the diseased tissues, we incubated aortic arches from arthritic mice fed WD with 1nM of RvT4. This concentration was chosen since in previous experiments we observed to regulate macrophage responses tested to the greatest extent (Figure 3A, B, D).”

With regards to the evaluation of this experiment using 2-way ANOVA, we carried out a repeated-measures, multivariate 2-way ANOVA on Graphpad Prism. We compared apoptotic cell uptake between the various concentrations of RvT4 at the 30 minute time point using a multiple comparisons test.

Figure 5 - The statistics is unclear. Panel G is a typical situation for 2-way ANOVA and the interactions, but 1-way ANOVA is mentioned at the end of the legend. This should be consulted with a statistician, appropriate tests should be used, and only the hypotheses tested by these tests should be discussed.

We thank the reviewer for bringing this to our attention, we agree with their comment and have revised the statistical analysis using a 2-way ANOVA.

Figure 6 - The same as above. Panel C should be evaluated using 2-way ANOVA, etc.

We thank the reviewer for bringing this to our attention. We have updated our statistical analysis to use a 2-way ANOVA in the mentioned figures 5A, 5B, 6C, and additionally figures 5C, 5D, 5E, 5F, 5G, 5I, 5J, 6B, and 6D. 2-way ANOVA results were corrected with the two-stage step-up- method of Benjamini, Krieger and Yekutieli for multiple comparisons by controlling the false discovery rate.

Minor points:

There are many typos in the new (yellow) text.

We thank the reviewer for bringing this to our attention. We have addressed these typos.

Figure 4B, please reformat the units

We have reformatted the units of this Figure 4B to Total cholesterol (ug/250,000 cells).

REVIEWER COMMENTS

Reviewer #1 (Remarks to the Author):

The authors have added in this re-revised ms the additional information requested, either with new experiments or by clarification of doubts. I have no further queries.

Reviewer #2 (Remarks to the Author):

The statistics is still incomplete. Please consult a professional statistician how to report the results. Figure 2 PCA – the loading plot is included, but the model parameters (e.g. Scree plot in Metaboanalyst) is missing. This defines how many principal components you need to interpret the data, etc.

For instance, Figure 5C. 2-way ANOVA was used, but where are the results? It seems that the factor “A” (RvT4 treatment) has a significant effect, but what about the siRNA factor? The 1st bar seems to be higher than the 3rd and than the 5th bar. Is that significant? Does the knockdown limit the lipid upload pathway? There are other questions like this throughout the manuscript.

I think that the statistics should be performed correctly and not only 'ad hoc' (pick comparisons of selected bars) to fit the narrative.

Reviewer #2 (Remarks to the Author):

We would like to thank the reviewer for their comments. In addition to adding n numbers to lipid mediator datasets presented in figure 1D and E we have also addressed the statistical queries raised in the latest round of comments. The responses to each of the comments are below and highlighted in the manuscript in yellow.

The statistics is still incomplete. Please consult a professional statistician how to report the results. Figure 2 PCA – the loading plot is included, but the model parameters (e.g. Scree plot in Metaboanalyst) is missing. This defines how many principal components you need to interpret the data, etc.

We include the corresponding Scree plots for results presented in Figure 2. These are presented in Supplemental Figure 4. We have also revised the figures to report the results of three principle components which together account for > 80% of the variation.

For instance, Figure 5C. 2-way ANOVA was used, but where are the results? It seems that the factor “A” (RvT4 treatment) has a significant effect, but what about the siRNA factor? The 1st bar seems to be higher than the 3rd and than the 5th bar. Is that significant? Does the knockdown limit the lipid upload pathway? There are other questions like this throughout the manuscript. I think that the statistics should be performed correctly and not only 'ad hoc' (pick comparisons of selected bars) to fit the narrative.

We would like to clarify that the statistical analysis was not performed 'ad hoc'. Rather we reported the results that were relevant to the hypothesis being tested, primarily due to space constraints and in the interest of clarity, especially since many of the other effects have been reported. Nonetheless, we revised the figures to report all the biologically relevant comparisons. The revised panels are found in Figures 5-7.

REVIEWER COMMENTS

Reviewer #2 (Remarks to the Author):

The authors updated the statistics and added several data points (animals) to the critical graphs.

However, the specificity of the methodology has recently been fundamentally questioned by several laboratories in the field.

<https://doi.org/10.3389/fphar.2022.838782>

https://zenodo.org/record/5766267#.Yrl5_OxByUk

(also related to rheumatoid arthritis)

The paragraph related to lipid mediator profiling has been updated:

“Lipid mediators were identified based on matching retention time of the peak of interest with those of authentic or synthetic standards and a signal to noise ratio > 5. Signal-to-noise ratios were calculated using the ‘Relative Noise’ algorithm.”

Although I do not know the parameters of ‘Relative noise’ algorithm, the illustrative panels F to I of Figure S1 are not very convincing. I would hardly assign a peak of RvT4 with S/N ratio > 5 at panel G. Similarly, the peak RvT4, Figure S4 seems to be shifted to RT ~13.9 min when compared to plasma Figure S1, panel G & I (RT ~13.6 min). It might be a matrix effect, but based on the relative peak heights and background noise (limited information due to RT window setting), the profile and compound identity is questionable.

It would be optimal if the authors could deposit the raw LC-MS data into a repository like Zenodo and include the data availability statement in the manuscript. The members of the lipid mediators field could do an independent evaluation of the peak values.

Response to reviewer comments

Reviewer #2 (Remarks to the Author):

The authors updated the statistics and added several data points (animals) to the critical graphs.

However, the specificity of the methodology has recently been fundamentally questioned by several laboratories in the field.

<https://doi.org/10.3389/fphar.2022.838782>

https://zenodo.org/record/5766267#.YrI5_OxByUk

(also related to rheumatoid arthritis)

We would like to bring to the reviewer's attention our detailed and extensive response to the allegations made against our methodologies which is published here:

<https://www.biorxiv.org/content/10.1101/2022.04.28.489064v1>

Amongst the points we address in our response are the following critical aspects:

- 1) The authors of the pre-print misrepresent a key aspect of the methodology that we employed whereby we specifically detail that a peak with a matching retention time to that of the relevant standard and not any region within the chromatogram is required to have an AUC >2000 units
- 2) The instrumentation used by the authors to collect MS/MS spectra appears to be contaminated since when we evaluated blank runs on our equipment, we did not observe the MS/MS spectra they present
- 3) The region where the MS/MS spectra they present was obtained was not specified and could have originated from anywhere within the chromatogram which would make the observation irrelevant
- 4) When we used a different methodologies, obtained essentially similar results.

In addition, many other laboratories have replicated findings made by our laboratory, as a testament to the robustness of our observations, some of these publications are cited in our response (see link above; these include articles from some of the Authors of the review you refer to), with many more available on PubMed.

We would also like to point out that apart from contravening fundamental principles of peer review, where the reviewers and the topic editor had a substantial conflict of interest and the reviewers have limited to no documented expertise in the field, the review cited above misreports and wrongly cites many of the findings reported in the literature. Furthermore, it selectively omits vast amounts literature from many laboratories, including that from the authors themselves, confirming identification and biology of SPM. Therefore, the relevance of the review to the work in our manuscript is questionable at best, if not biased and un-professional tout-court. In fact, we could cite other conflicts and below-par behavior behind the publication of this mediocre and biased review, but this is not relevant here.

An additional point to be noted is the fact that the methodologies that were employed in our published article are distinct from those employed in the present manuscript. Furthermore, we provide a side-by-side comparison of the methodologies employed herein to classical methodologies which demonstrates that the findings are essentially the same whether one uses the peak-to-peak methodology, or the relative noise methodology. Please see response below.

The paragraph related to lipid mediator profiling has been updated:

“Lipid mediators were identified based on matching retention time of the peak of interest with those of authentic or synthetic standards and a signal to noise ratio > 5. Signal-to-noise ratios were calculated using the ‘Relative Noise’ algorithm.”

Although I do not know the parameters of ‘Relative noise’ algorithm, the illustrative panels F to I of Figure S1 are not very convincing. I would hardly assign a peak of RvT4 with S/N ratio > 5 at panel G. Similarly, the peak RvT4, Figure S4 seems to be shifted to RT ~13.9 min when compared to plasma Figure S1, panel G & I (RT ~13.6 min). It might be a matrix effect, but based on the relative peak heights and background noise (limited information due to RT window setting), the profile and compound identity is questionable.

In support of the utility and robustness of the methodologies employed, we have provided a side-by-side comparison of the s/n values obtained using the peak-to-peak method and those obtained using the relative noise methodology for data presented in Figure 1D, E, F and G. As can be observed from the raw data presented in Appendix I, while there are differences in the s/n values obtained using the different methodologies the overall findings made would not change if we were to use the peak to peak methodology except for those made for $\Delta 6$ -trans-LTB₄ where - due to a number of closely eluting isomers - it is not possible to accurately calculate the s/n ratio, using the peak to peak methodology, since the chromatographic region around the peak of interest is occupied by these isomers. Indeed, this is one of the major limitations of the peak-to-peak methodology, whereby when biological isomers elute closely it is not possible to accurately calculate the s/n ratio and one must use a region which is usually far from the peak of interest. A second limitation remains the subjectivity of defining the noise region, and its location is likely to vary from sample to sample. Therefore, given that the fundamental findings of this study do not change and that the relative noise ratio methodology is an unbiased methodology that calculates the noise under the peak of interest, rather than in an arbitrarily defined region somewhere in the chromatogram, we have opted to retain the findings made using the relative noise methodology.

With regards to the point made by the reviewer on the slight shift in retention times between the data presented Figure S1 G&I and the one presented in Figure S4, we would like to highlight that the retention times tend vary slightly from run to run and from instrument to instrument. To account for this minor variation, we employ a standard mix that is utilized in each run, which contains both the mediator of interest, in this case RvT4 and a relevant deuterium labelled internal standard, in this case d₄-LTB₄. The latter standard is also present in each of the samples and used to establish the retention time of the mediator by determining the relative retention time of the deuterium label and the mediator (RvT4) in the standard mix. In Appendix II we provide the chromatograms for RvT4 and d₄-LTB₄ in the standard mix and in the samples for data obtained in Figure 1 and Figure S4

It would be optimal if the authors could deposit the raw LC-MS data into a repository like Zenodo and include the data availability statement in the manuscript. The members of the lipid mediators field could do an independent evaluation of the peak values.

We have uploaded the relevant raw data files to the EBI repository (EBI Accession number: S-BSST880)

Appendix 1

Chromatograms related to Figure 1D

PGD₂

Peak to Peak method

Relative Noise Method

Data highlighted in blue relates to the integrated peak in the chromatogram

Appendix 1

Chromatograms related to Figure 1D

PGD₂

Peak to Peak method

Relative Noise Method

Data highlighted in blue relates to the integrated peak in the chromatogram

Appendix 1

Chromatograms related to Figure 1D

PGD₂

Peak to Peak method

Data highlighted in blue relates to the integrated peak in the chromatogram

Appendix 1

Chromatograms related to Figure 1D

PGE₂

Peak to Peak method

Relative Noise Method

Sample Name	Sample Type	Component Name	Expected RT	Area	Retention Time	IS TI	Retention Time	Signal / Noise	Points Across...
QC	Unk...	PGE2 189	10.73	N/A	N/A	N/A	N/A	N/A	N/A
1-Wtchow...	Unk...	PGE2 189	10.51	1.138e6	10.51	N/A	0.00	8.1	24

Sample Name	Sample Type	Component Name	Expected RT	Area	Retention Time	IS TI	Retention Time	Signal / Noise	Points Across...
QC	Unk...	PGE2 189	10.55	4.994e2	10.55	N/A	0.00	7.4	12
1-Wtchow...	Unk...	PGE2 189	10.55	1.138e6	10.51	N/A	0.04	175.6	25
QC post	Unk...	PGE2 189	10.55	7.158e2	10.52	N/A	0.03	5.8	14

Sample Name	Sample Type	Component Name	Expected RT	Area	Retention Time	IS TI	Retention Time	Signal / Noise	Points Across...
QC post	Unk...	PGE2 189	10.73	N/A	N/A	N/A	N/A	N/A	N/A
5-Wtchow...	Unk...	PGE2 189	10.49	2.737e6	10.49	N/A	0.00	68.0	27

Sample Name	Sample Type	Component Name	Expected RT	Area	Retention Time	IS TI	Retention Time	Signal / Noise	Points Across...
5-Wtchow...	Unk...	PGE2 189	10.55	2.737e6	10.49	N/A	0.06	1891.2	27
6-Wtchow...	Unk...	PGE2 189	10.55	8.875e5	10.50	N/A	0.05	211.8	24

Data highlighted in blue relates to the integrated peak in the chromatogram

Appendix 1

Chromatograms related to Figure 1D

PGE₂

Peak to Peak method

Relative Noise Method

Data highlighted in blue relates to the integrated peak in the chromatogram

Appendix 1

Chromatograms related to Figure 1D

PGE₂

Peak to Peak method

Relative Noise Method

Data highlighted in blue relates to the integrated peak in the chromatogram

Appendix 1

Chromatograms related to Figure 1D

PGF_{2a}

Peak to Peak method

Relative Noise Method

Data highlighted in blue relates to the integrated peak in the chromatogram

Appendix 1

Chromatograms related to Figure 1D

PGF_{2a}

Peak to Peak method

Relative Noise Method

Data highlighted in blue relates to the integrated peak in the chromatogram

Appendix 1

Chromatograms related to Figure 1D

PGF_{2a}

Peak to Peak method

Relative Noise Method

Data highlighted in blue relates to the integrated peak in the chromatogram

Appendix 1

Chromatograms related to Figure 1D

LTB₄

Peak to Peak method

Relative Noise Method

Data highlighted in blue relates to the integrated peak in the chromatogram

Appendix 1

Chromatograms related to Figure 1D

LTB₄

Peak to Peak method

Relative Noise Method

Data highlighted in blue relates to the integrated peak in the chromatogram

Appendix 1

Chromatograms related to Figure 1D

LTB₄

Peak to Peak method

Relative Noise Method

Data highlighted in blue relates to the integrated peak in the chromatogram

Appendix 1

Chromatograms related to Figure 1D

D6-trans-LTB₄

Peak to Peak method

Relative Noise Method

Data highlighted in blue relates to the integrated peak in the chromatogram

Appendix 1

Chromatograms related to Figure 1D

D6-trans-LTB₄

Peak to Peak method

Relative Noise Method

Data highlighted in blue relates to the integrated peak in the chromatogram

Appendix 1

Chromatograms related to Figure 1D

D6-trans-LTB₄

Peak to Peak method

Relative Noise Method

Data highlighted in blue relates to the integrated peak in the chromatogram

Appendix 1

Chromatograms related to Figure 1D

D6-trans, 12-epi-LTB₄

Peak to Peak method

Relative Noise Method

Data highlighted in blue relates to the integrated peak in the chromatogram

Appendix 1

Chromatograms related to Figure 1D

D6-trans, 12-epi-LTB₄

Peak to Peak method

Relative Noise Method

Data highlighted in blue relates to the integrated peak in the chromatogram

Appendix 1

Chromatograms related to Figure 1D

D6-trans, 12-epi-LTB₄

Peak to Peak method

Relative Noise Method

Data highlighted in blue relates to the integrated peak in the chromatogram

Appendix 1

Chromatograms related to Figure 1E

RvT1

Peak to Peak method

Relative Noise Method

Sample Name	Sample Type	Component Name	Expected RT	Area	Retention Time	IS TI	Retention Time	Signal / Noise	Points Across
cross ref 2UL	Unk...	RvT1n3dpa 211	10.50	N/A	N/A	N/A	N/A	N/A	N/A
QC	Unk...	RvT1n3dpa 211	10.50	N/A	N/A	N/A	N/A	N/A	N/A
1-plasma...	Unk...	RvT1n3dpa 211	10.51	8.312e3	10.50	N/A	0.00	4.6	11

Sample Name	Sample Type	Component Name	Expected RT	Area	Retention Time	IS TI	Retention Time	Signal / Noise	Points Across
1-plasma...	Unk...	RvT1n3dpa 211	10.50	7.899e3	10.50	N/A	0.00	7.9	10
2-plasma...	Unk...	RvT1n3dpa 211	10.50	2.339e3	10.53	N/A	0.03	3.6	8
3-plasma...	Unk...	RvT1n3dpa 211	10.50	8.956e3	10.52	N/A	0.02	18.1	15

Sample Name	Sample Type	Component Name	Expected RT	Area	Retention Time	IS TI	Retention Time	Signal / Noise	Points Across
2-plasma...	Unk...	RvT1n3dpa 211	10.50	N/A	N/A	N/A	N/A	N/A	N/A
3-plasma...	Unk...	RvT1n3dpa 211	10.52	1.852e4	10.52	N/A	0.00	2.7	18
QC	Unk...	RvT1n3dpa 211	10.50	N/A	N/A	N/A	N/A	N/A	N/A

Sample Name	Sample Type	Component Name	Expected RT	Area	Retention Time	IS TI	Retention Time	Signal / Noise	Point Across
2-plasma Wfchow	Unk...	RvT1n3dpa 211	10.50	N/A	N/A	N/A	N/A	N/A	N/A
3-plasma Wfchow	Unk...	RvT1n3dpa 211	10.50	1.803e4	10.52	N/A	0.02	9.2	16

Data highlighted in blue relates to the integrated peak in the chromatogram

Appendix 1

Chromatograms related to Figure 1E

RvT1

Peak to Peak method

Relative Noise Method

Data highlighted in blue relates to the integrated peak in the chromatogram

Appendix 1

Chromatograms related to Figure 1E

RvT1

Peak to Peak method

Relative Noise Method

Data highlighted in blue relates to the integrated peak in the chromatogram

Appendix 1

Chromatograms related to Figure 1E

RvT4

Peak to Peak method

Relative Noise Method

Data highlighted in blue relates to the integrated peak in the chromatogram

Appendix 1

Chromatograms related to Figure 1E

RvT4

Peak to Peak method

Relative Noise Method

Data highlighted in blue relates to the integrated peak in the chromatogram

Appendix 1

Chromatograms related to Figure 1E

RvT4

Peak to Peak method

Relative Noise Method

Sample Name	Sa Typ	Component Name	Expected RT	Area	Retent... Time	IS Ti	Reten... Time...	Signal / Noise	Points Across...
2-plasma...	Unk...	RvT4n3dpa 211	13.70	1.512e4	13.63	N/A	0.07	5.0	12
3-plasma...	Unk...	RvT4n3dpa 211	13.57	4.930e3	13.64	N/A	0.07	6.8	9
QC	Unk...	RvT4n3dpa 211	13.70	N/A	N/A	N/A	N/A	N/A	N/A

Sample Name	Sa Typ	Component Name	Expected RT	Area	Retent... Time	IS Ti	Reten... Time...	Signal / Noise	Points Across...
1-plasma...	Unk...	RvT4n3dpa 211	13.70	9.789e3	13.64	N/A	0.06	8.0	12
2-plasma...	Unk...	RvT4n3dpa 211	13.64	1.335e4	13.64	N/A	0.00	15.7	14
3-plasma...	Unk...	RvT4n3dpa 211	13.70	4.949e3	13.64	N/A	0.06	6.8	10

Sample Name	Sa Typ	Component Name	Expected RT	Area	Retent... Time	IS Ti	Reten... Time...	Signal / Noise	Points Across...
4-plasma...	Unk...	RvT4n3dpa 211	13.70	N/A	N/A	N/A	N/A	N/A	N/A
5-plasma...	Unk...	RvT4n3dpa 211	13.70	N/A	N/A	N/A	N/A	N/A	N/A
6-plasma...	Unk...	RvT4n3dpa 211	13.70	6.107e3	13.64	N/A	0.06	5.3	9

Sample Name	Sa Typ	Component Name	Expected RT	Area	Retent... Time	IS Ti	Reten... Time...	Signal / Noise	Points Across...
4-plasma...	Unk...	RvT4n3dpa 211	13.70	N/A	N/A	N/A	N/A	N/A	N/A
5-plasma...	Unk...	RvT4n3dpa 211	13.70	N/A	N/A	N/A	N/A	N/A	N/A
6-plasma...	Unk...	RvT4n3dpa 211	13.70	5.642e3	13.64	N/A	0.06	8.4	7

Appendix 1

Chromatograms related to Figure 1F

RvT1

Peak to Peak method

Relative Noise Method

Appendix 1

Chromatograms related to Figure 1G

RvT4

Peak to Peak method

Relative Noise Method

Data highlighted in blue relates to the integrated peak in the chromatogram

Appendix 1

Chromatograms related to Figure 1G

RvT4

Peak to Peak method

Relative Noise Method

Sample Name	Sample Type	Component Name	Expected RT	Area	Retention Time	IS TI	Retention Time	Signal / Noise	Points Across
plasma 6	Unk...	RvT4n3dpa 211	14.05	N/A	N/A	N/A	N/A	N/A	N/A
plasma 7	Unk...	RvT4n3dpa 211	14.05	4.101e3	13.66	N/A	0.39	5.0	10
plasma 9	Unk...	RvT4n3dpa 211	14.05	N/A	N/A	N/A	N/A	N/A	N/A

Sample Name	Sample Type	Component Name	Expected RT	Area	Retention Time	IS TI	Retention Time	Signal / Noise	Points Across
plasma 6	Unk...	RvT4n3dpa 211	13.70	N/A	N/A	N/A	N/A	N/A	N/A
plasma 7	Unk...	RvT4n3dpa 211	13.70	2.565e3	13.67	N/A	0.03	8.6	9
plasma 9	Unk...	RvT4n3dpa 211	13.70	N/A	N/A	N/A	N/A	N/A	N/A

Sample Name	Sample Type	Component Name	Expected RT	Area	Retention Time	IS TI	Retention Time	Signal / Noise	Points Across
plasma 10	Unk...	RvT4n3dpa 211	14.05	N/A	N/A	N/A	N/A	N/A	N/A
plasma 14	Unk...	RvT4n3dpa 211	13.68	4.551e3	13.69	N/A	0.01	4.3	9
plasma 15	Unk...	RvT4n3dpa 211	14.05	N/A	N/A	N/A	N/A	N/A	N/A

Sample Name	Sample Type	Component Name	Expected RT	Area	Retention Time	IS TI	Retention Time	Signal / Noise	Points Across
plasma 10	Unk...	RvT4n3dpa 211	13.70	N/A	N/A	N/A	N/A	N/A	N/A
plasma 14	Unk...	RvT4n3dpa 211	13.70	3.300e3	13.69	N/A	0.01	12.7	10
plasma 15	Unk...	RvT4n3dpa 211	13.70	N/A	N/A	N/A	N/A	N/A	N/A

Appendix 1

Chromatograms related to Figure 1G

RvT4

Peak to Peak method

Relative Noise Method

Appendix 2

Chromatograms related to Figure S1E

RvT4 in Std Mix

d₄-LTB₄ in Std Mix

RvT4 in Sample

d₄-LTB₄ in Sample

Data highlighted in blue relates to the integrated peak in the chromatogram

RvT4 in Std Mix

d₄-LTB₄ in Std Mix

RvT4 in Sample

d₄-LTB₄ in Sample

REVIEWER COMMENTS

Reviewer #2 (Remarks to the Author):

I have no further questions.

Reviewer #4 (Remarks to the Author):

Within this paper 'Resolvin T4 promotes reverse cholesterol transport in lipid laden macrophages to reduce vascular inflammation', Walker et al., firstly demonstrate and inverse relationship between RvT4 plasma levels in mice and degree of atherosclerosis in arthritic mice fed an atherogenic diet. The authors progress to demonstrate that administration of RvT4 to the mice could reduce atherosclerotic burden within the mice. The mechanisms are further explored (primarily ex vivo) whereby the authors suggest that promotion of reverse cholesterol transport (RCT) is the key mechanism underpinning beneficial effects – with a particular emphasis on a protective beneficial effect on macrophage efflux capacity.

Comments:

This paper has demonstrated an inverse relationship between RvT4 levels and atherosclerosis in an arthritic mouse model with n=12 which formed the basis of the subsequent experiments (it is not clear whether these n=12 are WT+ApoE combined +/- chow diet +/- western diet?). The abstract alludes to these results 'Herein, we found that plasma concentrations of the pro-resolving mediator series resolvin (RvT)4 were negatively correlated with vascular lipid load in inflammatory arthritis' – it should be clearly stated here that this was in a preclinical study. It would have greatly strengthened the paper if the authors could similarly demonstrate such an association in humans.

The paper's title states that 'Resolving T4 promotes reverse cholesterol transport' but after reviewing the manuscript I have serious concerns about this statement. The results at best demonstrate that RvT4 attenuates atherosclerosis in an arthritic model that is associated with increased capacity to support cellular cholesterol efflux (via SRBI). Causation has not been demonstrated in vivo that modulation of macrophage SR-BI is the primary mechanism underpinning beneficial effects of RvT4 in vivo.

Reverse cholesterol transport (RCT) is an extremely complex pathway involving cholesterol flux from peripheral cells onto circulating HDL particles to be returned to the liver for trafficking into bile/feces. While it is quite plausible that RvT4 could have reduced inflammation in the arthritis model to improve inflammatory-impaired RCT, this was not evaluated by the authors. Indeed, in the arthritic model itself there is no investigation of macrophage-to-feces RCT which would have been extremely valuable. In figure 6G&H the authors start to address this angle but with extremely low n-numbers (major concern throughout) and only in ApoE^{-/-} mice (presuming no arthritis in this model?). While SR-BI levels were monitored in the liver cells the n-number is just too low to infer any meaningful results for such a study. No measure of HDL function was assessed throughout – HDL efflux capacity is significantly impaired under inflammatory conditions independent of HDL-cholesterol levels. It was a missed opportunity not to have evaluated this important step of RCT in the arthritic mouse model.

Another major concern is the lack of control in efflux assays. Addition of most lipids into the supernatant will inherently stimulate some basal cholesterol flux out of the cell (also observed if you add LPS to the cells – RvT1 would have been an ideal control for these studies). This would explain the rapid efflux of cholesterol from the cells. It would have been very interesting to establish whether RvT4 could attenuate inflammatory-impaired cholesterol flux (which would have been more relevant to their in vivo model) but this was apparently not explored. It would also be helpful if the authors could clarify for efflux assays as to whether all experiments were conducted with DiI-labelled oxLDL – the figure legends suggest that the cells were just labelled with ox-LDL and free cholesterol measured in the supernatant? But the methods section suggests the method involves fluorescence measurements. Please clarify as there is a major difference between these two potential methods with measurement of free cholesterol in the media being particularly crude. Did the researchers use minimal essential media for these efflux assays? Please clarify the media used and whether a minimal essential media control was used to subtract for basal efflux from the cells. The purpose of the experiments on F4/80+ cells were not clear in Figure 4D&E – (n-number here is particularly problematic). This to me does not add to the story. Finally throughout the manuscript the manipulation of the Y-axis to enhance the appearance of results was problematic for me.

Specific comments below:

Figure 1: I am very concerned how the study was performed in n=7 mice but measurements of arthritis and aortic lipid levels was performed in n=3 mice? What happened to the other n=4 mice? This low-n number is extremely problematic and no evidence of reproducibility of effect has been demonstrated. The starting number of n=7 mice was itself particularly low. Notwithstanding the low n-number, the effects of RvT4 on atherosclerosis lesion burden is the major strength of the paper.

Please include your r-value and give more information about the source of the data for the n=12 used in the correlation.

Figure 2: Gene expression performed in n=3 mice even though there were n=7 mice per group.

Figure 3: No major comment here but seemed a little out of place for this paper and was apparently not investigated any further.

Figure 4: As per comments at the start I have concerns over these efflux assays and lack of a lipid control. Clarification around the media needs to be addressed. The alteration of the Y-axis is particularly problematic. And Figure 4D-E do not add much weight to the story (n-number too low to infer anything). Also when you did this study did you inject the RvT4 into the same intraperitoneal site and if so what was the difference in timing between injecting BODIBI-FL-cholesterol and injecting RvT4?

Figure 5: Again manipulation of the Y-axis was problematic here. For all the knock-down studies the protein expression of the transporters should be included to demonstrate knock-down efficiency – please include this data in the supplement (apologies if I missed it).

Figure 6: Please correct the axes throughout this figure. The translocation of SRBI is extremely questionable within these experiments and effect size very small (made larger by the manipulation of the axis – again n=3 is very low). Same comments for Figure 6C&D – to me there is no difference in macrophage neutral lipid load. In Figure 6F the authors measured HDL-associated cholesterol after FPLC profiling (in the arthritic mouse model). This is probably one of the most important results of the study but buried in this figure. This data should be presented as FPLC curves with cholesterol mass measurements (not ratios). Again however, the n=3 is extremely low here for FPLC analysis (we would typically use n=16 mice as the variability is very high). All mice should have been analysed from the initial studies for their FPLC profiles.

Figure 6G&H start to address the latter steps of RCT but this was just done in ApoE^{-/-} mice with no arthritis. This is a major limitation. Very little can be inferred from the findings here and the study has been massively underpowered for RCT studies.

Overall this study has potential but there is a sense that the authors have spread themselves too thin during the analysis of the data – some experiments are not adding to the story and the n-numbers are too low to overly interpret the findings. The exclusion of post-mortem analysis of mice that were included within the in vivo studies is particularly problematic. The paper would have benefited on more focus on RCT with all mice included in the subsequent analysis.

Reviewer #5 (Remarks to the Author):

Walker ME et al provide compelling evidence that RvT4 may be associated with arthritis-exacerbated atherosclerosis. The authors found a decrease in RvT4, then took careful consideration for understanding causation by adding back RvT4 exogenously. Moreover, the authors identified that one mechanism associated with RvT4's protective actions are via promoting reverse cholesterol transport with SR-B1.

Positive aspects:

-This combo model is exciting and atherosclerosis in humans is rarely “only” atherosclerosis. I think combination models like the one presented here should be a new norm to study specific aspects of atherosclerosis. Therefore, the overall design is intriguing and clinically relevant.

-Identification that a decrease in a pro-resolving ligand is associated with advancing atherosclerosis is highly reproducible and is consistent with the literature (Fredman G et al 2016; Thul S et al 2017; Bazan HA et al 2018; Welty F et al FASEB). The novelty of this manuscript is that it introduces RvT4 as a new player. Along these lines, This study a) identifies RvT4 as a new mediator that may be important in limiting arthritis-exacerbated atherosclerosis b) provides a new mechanism for its actions. Understanding how these pro-resolving ligands work from a mechanistic perspective is important because they may have similar overall functions, but their mechanisms of action are probably quite distinct.

-The use of JetPei to test causation in vivo was intriguing and may be helpful to other athero researchers as a “quicker” method to test causation of a particular target.

-This manuscript was written clearly and logically

Points of clarification:

--This is ultimately the choice of the authors, but I don't think the title reflects the arthritis and atherosclerosis aspects. Perhaps instead of vascular inflammation, you can be more specific. Along these lines, is there a reason why the authors mention “vascular protective effects” and “vascular inflammation” when they could say atherosclerotic plaques or atherosclerosis or protective actions on plaques?

-Fig 3. F – can you please be clearer about the methods for this panel? This is not a “conventional” assay and a deeper method as to how you quantified monocytes that “migrate from” the aorta is requested.

-Fig. 5. Fig A,B,C,E,H and parts of Fig. 6-- the changes are statistically significant but numerically, not very different. Is this expected for this type of assay? Meaning are these small changes expected as part of this assay?

-Fig 7. Was RvT4 still able to decrease necrotic area in the SR-B1 KD experiment? Also, did RvT4 decrease plaque necrosis simply because the lesions were smaller? What is the % necrosis per lesion area?

REVIEWER COMMENTS

Reviewer #4 (Remarks to the Author):

This is a well written paper addressing a very important clinical problem and with an exciting new therapeutic. However, I stand over my original comments that the sample size is a major concern from my perspective.

It would have been important to see some simple additional experiments eg. to use RvT1 as a control in efflux assays as suggested. In any in vitro study with lipids it is always critical to have a lipid control - if RvT1 did the same thing as RvT4, then why was there no association to CVD with the former? This was the response to these important critiques 'These observations do not exclude that other lipid mediators, including RvT1, may regulate a similar pathway therefore we believe that the appropriate controls for the experiments conducted are those employed (i.e. vehicle controls) rather than another mediator/lipid'. This to me was a missed opportunity that would have greatly enhanced the quality of the paper.

It was also a weakness that the group started with an interesting model (arthritis) but then moved into ApoE^{-/-} mice and mixed all the results together saying the ApoE^{-/-} model is an inflammatory model. They are two very different disease states and inferring results from one model to the other model is just not possible. A validation study in ApoE^{-/-} mice may have been warranted but at the moment it is confusing to the reader – the logic behind switching models is not evident.

I think figure 7 is the strongest figure and they indeed did go back to the arthritis model here. It would be good to see the FPLC profiles from these mice to see how the treatment/knock-down ultimately impacted overall lipoprotein profiles in these animals.

But overall I have read over the replies and I unfortunately am left underwhelmed by the response. I think we need to make a concerted effort in the field to improve our issues with reproducibility - relying on other papers that got published with n=3/4 per group does not to me justify the use of such small numbers in this study. 'We would like to note that the n

numbers used in our studies are in line with those published by others (PMID: 29739755).¹ I can see now that this was picked up in previous rounds of review and additional mice added explaining the n=7 in some places and n=3 in others but to me the over-interpretation of results with n=3/4 needs to be cautioned.

While I think the fundamental findings of this study are interesting and important I think the authors have over-stretched focusing on quantity of results rather than quality which significantly impacts confidence within the results

Reviewer #5 (Remarks to the Author):

The authors addressed my comments.

Reviewer #4 (Remarks to the Author):

This is a well written paper addressing a very important clinical problem and with an exciting new therapeutic. However, I stand over my original comments that the sample size is a major concern from my perspective.

Q. It would have been important to see some simple additional experiments eg. to use RvT1 as a control in efflux assays as suggested. In any in vitro study with lipids it is always critical to have a lipid control - if RvT1 did the same thing as RvT4, then why was there no association to CVD with the former? This was the response to these important critiques 'These observations do not exclude that other lipid mediators, including RvT1, may regulate a similar pathway therefore we believe that the appropriate controls for the experiments conducted are those employed (i.e. vehicle controls) rather than another mediator/lipid'. This to me was a missed opportunity that would have greatly enhanced the quality of the paper.

R. We thank the reviewer for their comment. We have performed additional experiments comparing the ability of another lipid mediator we found to be regulated in mice fed a western diet. In these experiments we explored the ability of PGE₂, a lipid mediator that was upregulated in peripheral blood of mice fed a western diet. Here we observed that RvT4 is significantly more effective (~x2) at decreasing intracellular cholesterol level than PGE₂. The results are presented in Figure 4D and discussed on page 8 lines 207-210 as detailed below.

We next evaluated whether this ability of RvT4 to reduce intracellular cholesterol load was shared with other lipid mediators. Incubation of lipid loaded macrophages with PGE₂, a lipid mediator found to be upregulated in mice fed a WD, led to a decrease in intracellular lipid load, although this was significantly less than that induced by RvT4 (Figure 4D).

Q. It was also a weakness that the group started with an interesting model (arthritis) but then moved into ApoE^{-/-} mice and mixed all the results together saying the ApoE^{-/-} model is an inflammatory model. They are two very different disease states and inferring results from one model to the other model is just not possible. A validation study in ApoE^{-/-} mice may have been warranted but at the moment it is confusing to the reader – the logic behind switching models is not evident.

R. We agree that the metabolic dysfunction observed in the ApoE^{-/-} model is a different disease state to arthritis, it is also important to note that in many patients these two disease states co-exist. The use of ApoE^{-/-} mice is linked to the aim of the study to establish the relevance of cholesterol efflux in macrophages to protection from the development of atherosclerosis, a common co-morbidity in patients with arthritis. As WT mice are resistant to development of atherosclerotic lesions, we could not explore this aspect of pathology using WT arthritic mice. For this purpose, we opted to employ ApoE^{-/-} since the metabolic dysfunction observed in these animals is also observed in many patients with inflammatory arthritis and these animals develop atherosclerotic lesions. We revised the text in the manuscript on page 10 lines 272-279 to clarify the rationale as detailed below:

RA patients often present with metabolic dysfunction that is thought to exacerbate vascular disease³⁴. Since WT mice are resistant to the development of atherosclerotic lesions, to test whether RvT4 regulates macrophage cholesterol efflux in vivo in settings relevant to the human RA we employed the ApoE^{-/-} model that develops metabolic dysfunction and advanced atherosclerosis and combined this K/BxN serum transfer arthritis. Administration of RvT4 to

ApoE^{-/-} arthritic mice fed WD led to an overall decrease in lipoprotein-associated cholesterol in plasma and an increase in the ratio of HDL-associated cholesterol to total cholesterol (Figure 6F,G).

Q. I think figure 7 is the strongest figure and they indeed did go back to the arthritis model here. It would be good to see the FPLC profiles from these mice to see how the treatment/knock-down ultimately impacted overall lipoprotein profiles in these animals.

R. We would like to point out that results presented in Figure 6 also present results from *ApoE^{-/-}* mice with arthritis.

Results for cholesterol associated lipoprotein levels presented in Figure 7 were obtained using commercially available kits rather than FPLC as at the time these experiments were performed (Dec 2020-Feb 2021) we did not have access to the FPLC system due to ongoing restrictions related to the COVID-19 pandemic. We have revised the methods section to clarify this as detailed on page 33 lines 1051-1055 and detailed below:

In experiments presented in Figure 7 HDL and LDL/VLDL associated cholesterol were determined using the HDL and LDL/VLDL Cholesterol Assay Kit (Abcam, #ab65390) following manufacturer's instruction.

Q. But overall I have read over the replies and I unfortunately am left underwhelmed by the response. I think we need to make a concerted effort in the field to improve our issues with reproducibility - relying on other papers that got published with n=3/4 per group does not to me justify the use of such small numbers in this study. 'We would like to note that the n numbers used in our studies are in line with those published by others (PMID: 29739755).' I can see now that this was picked up in previous rounds of review and additional mice added explaining the n=7 in some places and n=3 in others but to me the over-interpretation of results with n=3/4 needs to be cautioned.

R. We performed additional experiments to increase the n numbers for experiments in Figure 6. We would also like to iterate that results presented in our manuscript are 1) derived from at least two independent experiments and 2) supported by power calculations. The fact that these results were obtained from two independent experiments, in many cases with replicates performed years apart, supports the robustness of our observations.

Q. While I think the fundamental findings of this study are interesting and important I think the authors have over-stretched focusing on quantity of results rather than quality which significantly impacts confidence within the results.

R. We thank the reviewer for their time in reviewing our manuscript and for their positive comments related to relevance of our fundamental observations. We trust that they are now also reassured on the quality of our observations.

REVIEWERS' COMMENTS

Reviewer #4 (Remarks to the Author):

There are additional experiments performed on this second round of revisions which have strengthened the paper in part. T

I have no further comments on the manuscript.

There are additional experiments performed on this second round of revisions which have strengthened the paper in part. T
I have no further comments on the manuscript.

We thank the reviewer for their comments which help strengthen our manuscript